# Polarity of the CRISPR roadblock to transcription

Porter M. Hall [1], James T. Inman[2,3], Robert M. Fulbright[2], Tung T. Le[2,3], Joshua J. Brewer[4], Guillaume Lambert [5], Seth A. Darst [4] & Michelle D. Wang [2,3] ✉

CRISPR (clustered regularly interspaced short palindromic repeats) utility relies on a stable Cas effector complex binding to its target site. However, a Cas complex bound to DNA may be removed by motor proteins carrying out host processes and the mechanism governing this removal remains unclear. Intriguingly, during CRISPR interference, RNA polymerase (RNAP) progression is only fully blocked by a bound endonuclease-deficient Cas (dCas) from the protospacer adjacent motif (PAM)-proximal side. By mapping dCas-DNA interactions at high resolution, we discovered that the collapse of the dCas R-loop allows *Escherichia coli* RNAP read-through from the PAM-distal side for both Sp–dCas9 and As–dCas12a. This finding is not unique to RNAP and holds for the Mfd translocase. This mechanistic understanding allowed us to modulate the dCas R-loop stability by modifying the guide RNAs. This work highlights the importance of the R-loop in dCas-binding stability and provides valuable mechanistic insights for broad applications of CRISPR technology.

The utilization of CRISPR-associated (Cas) nucleases offers the ability to precisely target DNA sequences and cleave at those sites, enabling great advances in gene editing, targeting and diagnostic technology for both prokaryotic and eukaryotic systems[1–4]. To accomplish this, a Cas protein is complexed with a guide RNA (gRNA) that contains a spacer region complementary to the target DNA sequence. A critical facet of CRISPR utility relies on Cas enzyme-binding stability, which is dictated by specific and robust binding of the gRNA to the target DNA sequence. This occurs via recognition of a PAM sequence and hybridization of the spacer region of the gRNA with the target DNA to form a gRNA-DNA hybrid (R-loop)[1,5].

In vivo, a DNA-bound Cas can not only dissociate from the DNA spontaneously but also be removed by motor proteins carrying out other host processes. However, the mechanism governing Cas removal by motor proteins is not well understood. Intriguingly, during CRISPR interference (CRISPRi), which uses a dCas to block transcription, the effectiveness of dCas removal depends on the orientation of the bound dCas relative to transcription. Transcription elongation is rather permissive from the PAM-distal side of a bound dCas but is predominantly

blocked from the PAM-proximal side[6–8]. Curiously, a bound dCas is not found to be a polar barrier to replication[9,10], indicating that the polarity is dictated by the dynamics of how motor proteins overcome dCas barriers. The CRISPRi system offers an appealing opportunity to examine the polarity and mechanics of dCas removal and, more broadly, Cas-binding stability.

Using single-molecule assays, we mapped the structural features of a dCas complex bound to DNA and investigated how an elongating RNAP interacts with the bound dCas. Through this, we discovered the mechanism for CRISPRi polarity and dCas removal, highlighting the importance of the R-loop stability for a bound Cas. This mechanistic understanding suggests strategies for modulating dCas stability and holds broader implications for Cas applications.

## Results

### R-loop of a dCas complex bound to DNA

To investigate the structural features that may underlie the polar barrier of a bound dCas, we first mapped protein-nucleic acid interactions of a

[1]Biophysics Program, Cornell University, Ithaca, NY, USA. [2]Department of Physics, Laboratory of Atomic and Solid State Physics, Cornell University, Ithaca, NY, USA. [3]Howard Hughes Medical Institute, Cornell University, Ithaca, NY, USA. [4]Laboratory of Molecular Biophysics, Rockefeller University, New York, NY, USA. [5]Department of Applied and Engineering Physics, Cornell University, Ithaca, NY, USA. ✉e-mail: mdw17@cornell.edu

bound dCas via a high-resolution 'DNA unzipping mapper' technique[11–14] (Fig. 1a). Using an optical trap, we unzipped DNA by mechanical separation of the two strands of a double-stranded DNA (dsDNA) containing a bound dCas. Before the unzipping fork encountered the bound dCas, the unzipping force followed the force signature of the corresponding naked DNA baseline, but, when the unzipping fork encountered the complex, the unzipping force deviated from the naked DNA baseline, indicating DNA interactions with dCas. To examine interaction polarity, we unzipped the DNA through a bound complex from either the PAM-distal side or the PAM-proximal side.

Using the unzipping mapper, we compared the force signatures of both a dCas9 and a dCas12a, two of the most prevalent Cas proteins (Fig. 1b). Although the target sequence of gRNA is located at the 5'-end for dCas9 and at the 3'-end for dCas12a, we found that their interaction maps were strikingly similar.

When unzipped from the PAM-distal side, both dCas complexes showed a drop in force below the naked DNA baseline, followed by a rise in force above the baseline. The drop in force is consistent with the presence of the gRNA-DNA hybrid, which prevents DNA base pairing, creates a DNA bubble and thus reduces the unzipping force. Note that, due to thermal DNA 'breathing' fluctuations, the unzipping fork detects the DNA bubble downstream[15], leading to an earlier drop in force. The drop in force indicates a lack of strong interactions between the dCas protein and DNA before the bubble. For dCas9, the subsequent rise in force was detected within the gRNA-DNA hybrid region, indicating strong interactions between dCas9 and DNA in that region. For dCas12a, two types of traces were detected (middle panel of Fig. 1b), 43% of the 37 traces measured show a single rise in force above the naked DNA force baseline within the gRNA-DNA hybrid region and the remaining traces show an additional rise in force above the naked DNA force baseline at the distal end of the gRNA-DNA hybrid. Both types of traces show a dip in force below the DNA baseline within the gRNA-DNA hybrid. These observations indicate that, although dCas9 assumes one dominant conformation, dCas12a may adopt two distinct conformations, as has been suggested by previous biochemical studies[16–20]. In contrast, when either a dCas9 or a dCas12a was unzipped from the PAM-proximal side, the force rose sharply at ~6 bp from the PAM site, indicating tight binding of the dCas protein to DNA at that region. The locations of this tight binding site are consistent with those suggested by structures of these complexes[21–23].

It is interesting that these force features bear a remarkable resemblance to those of an *E. coli* transcription elongation complex (TEC), which the DNA unzipping mapper method previously characterized[24–26]. For ease of direct comparison of data with dCas complexes, we remapped the TEC under the same experimental conditions as for the dCas complexes (Fig. 1b). When unzipped from upstream of transcription, a TEC showed a drop in force due to the transcription bubble containing the RNA-DNA hybrid, followed by a rise in force near the active site. When unzipped from downstream of transcription, a TEC showed a rise in force at 10–20 bp downstream of the active site, indicating tight binding of RNAP to DNA downstream of its active site.

### Hypothesized mechanism of dCas roadblock polarity

The unzipping mapper data (Fig. 1b) clearly demonstrate that, just like a TEC, a DNA-bound dCas complex contains an unprotected R-loop-mediated DNA bubble near one end and tightly clamped DNA at the other end. These shared structural features suggest that these complexes may be removed by a common mechanism. Previous bulk transcription studies showed that collapse of the transcription bubble leads to the destabilization of a TEC[27–29]. Thus, we speculate that a DNA-bound dCas may be destabilized similarly via DNA bubble collapse of a bound dCas.

This has led us to hypothesize the following mechanism for the polarity of CRISPRi (we reason that this polarity is inherent to the common structural features of TEC and dCas complexes): when a translocating RNAP approaches a bound dCas from the PAM-distal side (Fig. 1c), RNAP first encounters the DNA bubble of the dCas complex. As RNAP tightly clamps its downstream DNA, forward translocation will rezip the DNA bubble of the dCas complex. This leads to collapse of the DNA bubble of the dCas complex, disruption of the gRNA-DNA hybrid and ultimately removal of dCas from DNA. Thus, transcription from the PAM-distal side is likely to be more permissive. On the other hand, when RNAP approaches a bound dCas from the PAM-proximal side, RNAP will encounter a dCas roadblock that may be too strong for the RNAP to overcome (Extended Data Fig. 1). Thus, transcription from the PAM-proximal side is more prohibitive.

### A bound dCas is a highly asymmetrical roadblock

To test this hypothesis, we developed a single-molecule assay using the DNA unzipping mapper that quantitatively measures the ability of RNAP to transcribe through a bound dCas from either the PAM-distal side or the PAM-proximal side. In this assay (Fig. 2a), a DNA template initially contained a TEC paused at the A20 position via nucleotide starvation and a bound dCas downstream. A control experiment was conducted using the unzipping mapper to determine the occupancies of RNAP and dCas, which were both found to be >90%. Subsequently, transcription was resumed by the introduction of NTPs into the sample chamber and was then quenched after 135 s, which should have been sufficient time to allow most RNAPs to reach the bound dCas while limiting spontaneous dCas9 dissociation (Extended Data Fig. 2). Subsequently, the locations of bound proteins were detected using the unzipping mapper.

Unzipping traces taken after the NTP chase fell into several categories due to asynchronization of the RNAP population as a result of the stochastic nature of RNAP motion (Extended Data Fig. 3). Figure 2b shows representative traces of this assay when transcription approached dCas9 from the PAM-distal side. One example shows that a force peak was detected immediately before the dCas9 force peak, suggesting that RNAP stalled after colliding with dCas9 but was unable to remove dCas9. Another example trace shows that the only detected bound protein was downstream of dCas9, possibly due to RNAP having elongated forward after removing dCas9, but not having reached the template end. In contrast, when RNAP encountered dCas9 from the PAM-proximal end, most traces showed two rises in force at around 30 bp and 6 bp, respectively, before the PAM site (Extended Data Figs. 3 and 4b), consistent with RNAP stalling after collision with dCas9 but unable to remove dCas9. We also carried out a similar experiment to examine transcription through dCas12a (Extended Data Figs. 3 and 4c,d) and obtained a similar result.

These traces show very different transcription behaviors between the PAM-distal and PAM-proximal collisions and demonstrate that a bound dCas is a polar barrier to transcription. To accurately determine

---

**Fig. 1 | High-resolution maps of dCas interactions with DNA using the DNA unzipping mapper. a**, DNA unzipping mapper configuration. An unzipping template is tethered at one end to the surface of a coverslip of a sample chamber and at the other end to a polystyrene bead held in an optical trap. Using the optical trap, the bead is moved relative to the surface, progressively unzipping the DNA until the unzipping fork reaches a bound protein, which resists unzipping, leading to a distinct rise in force. The location of the rise in force is used to map the protein location. **b**, Representative unzipping traces (red) of bound dCas9 (top), bound dCas12a (middle) and a paused TEC (bottom), along with naked DNA traces (black). The DNA was unzipped from either direction (black arrows) relative to the bound protein for each protein. Two conformations were detected when a bound dCas12a protein was unzipped from the PAM-distal side, shown as light blue and red. The two dashed lines bracket the expected gRNA-DNA hybrid locations for dCas9 or dCas12a and the expected RNA-DNA hybrid of a TEC. The red arrows indicate locations where the unzipping force dipped below the naked DNA baseline. **c**, Hypothesized mechanism for transcription read-through from the PAM-distal side. Note that gRNA hybridizes with the TEC template and nontemplate strand for a bound dCas9 and dCas12a complex, respectively. Source data containing traces for **b** are provided.

transcription read-though from each side of a bound dCas, we carried out several control experiments to obtain the probability of a template initially not having a bound RNAP or dCas protein (Supplementary Table 1) and the probabilities of spontaneous dissociation of RNAP or dCas (Extended Data Fig. 2c,e). These probabilities were taken into account in the final read-through analysis (Extended Data Fig. 5).

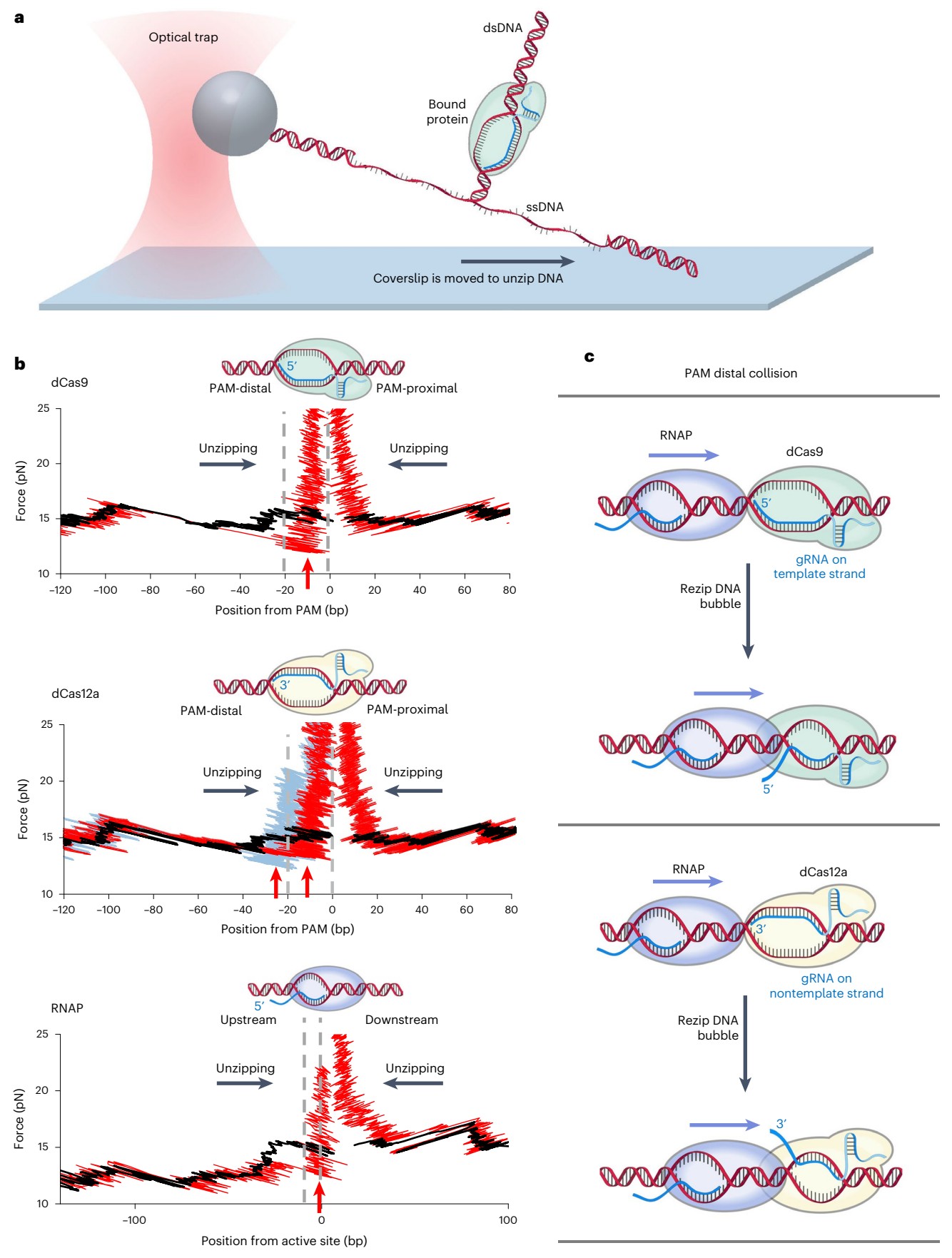

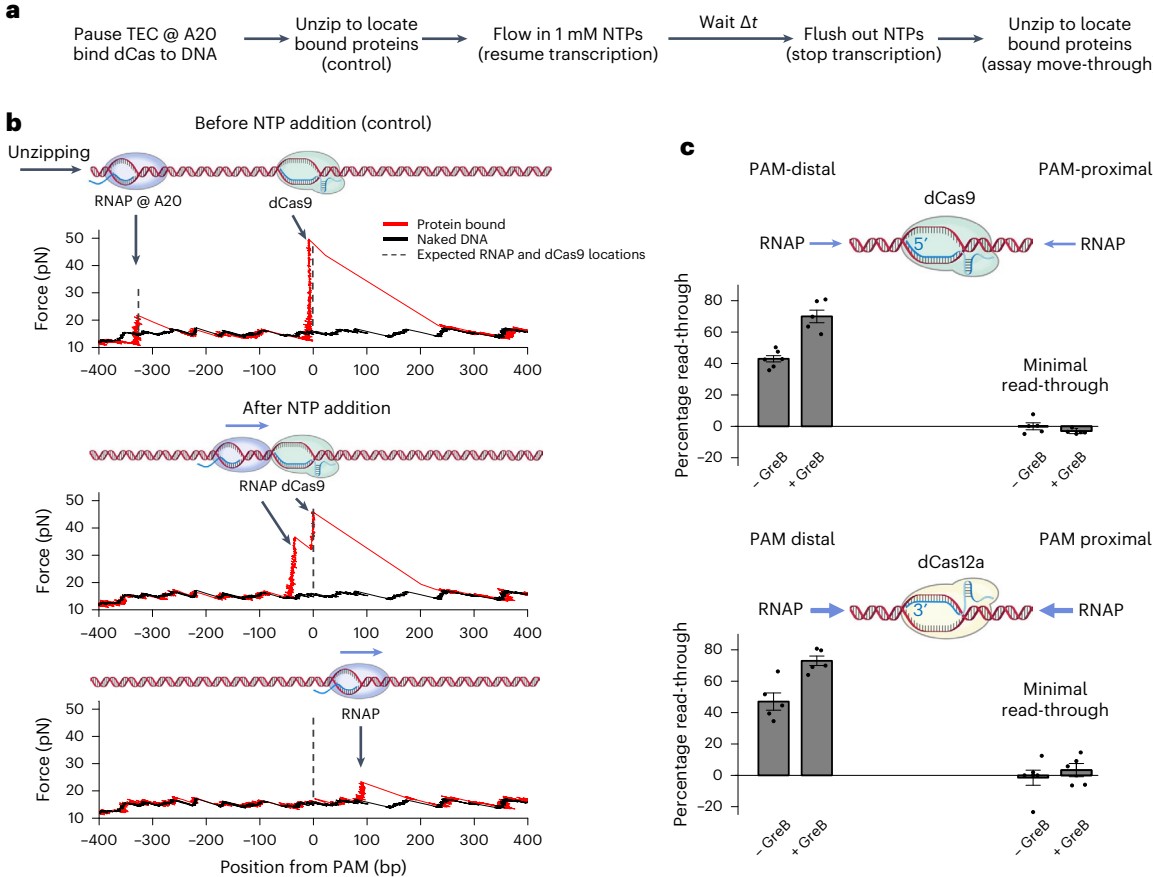

**Fig. 2 | A quantitative assay for transcription read-through of a bound dCas complex. a**, Flowchart of a single-molecule transcription assay for a given sample chamber. Some DNA tethers were used as controls to assay bound protein locations before NTP addition. Other tethers were used to assay bound protein locations after an NTP chase time of Δt = 135 s. DNA was always unzipped in the same direction as RNAP translocation. **b**, Representative traces of RNAP encountering a bound dCas9 from the PAM-distal side. An example control trace is shown with RNAP and bound dCas9 detected at their expected locations. After NTP addition, example traces of RNAP are shown colliding with dCas9 and reading through dCas9. Naked DNA traces are shown in black. **c**, Transcription read-through efficiency for RNAP encountering a bound dCas from either the

PAM-distal side or the PAM-proximal side. Results from both dCas9 (top) and dCas12a (bottom) are shown. For each sample chamber, both control traces and noncontrol traces were taken to obtain the read-through efficiency for that chamber. Each type of experiment was repeated using *n* biologically independent sample chambers: dCas9 PAM-distal, *n* = 6 (−GreB) and *n* = 6 (+GreB); dCas9 PAM-proximal, *n* = 5 (−GreB) and *n* = 5 (+GreB); dCas12a PAM-distal, *n* = 5 (−GreB) and *n* = 5 (+GreB); dCas12a PAM-proximal, *n* = 6 (−GreB) and *n* = 5 (+GreB). Read-through values were calculated for each sample chamber (black dots) and the mean and s.e.m. of these repeats are also shown. Source data containing traces for **b** and transcription read-through values for **c** are provided.

Using this method, we found that transcription read-through of a bound dCas9 showed an efficiency of 43% when RNAP approached dCas9 from the PAM-distal side and was undetectable from the PAM-proximal side (Fig. 2c). For dCas12a, the read-through efficiencies were similar to those for dCas9: 47% when RNAP approached dCas12a from the PAM-distal side and undetectable from the PAM-proximal side (Fig. 2c). To further validate these results from single-molecule studies, we carried out corresponding bulk transcription assays and the bulk data show a similar extent of polarity (Extended Data Fig. 6). These findings on the polarity of both the dCas9 and the dCas12a barriers to transcription are consistent with those from previous in vivo studies[6–8,30,31], while also providing a highly quantitative and controlled measure of the polarity.

When RNAP encountered a bound dCas but could not read through it, RNAP probably backtracked[24,32–34], where RNAP reverse translocates along DNA with its catalytic site disengaged from the 3′-end of the RNA, rendering transcription inactive[35,36]. *E. coli* GreB is a transcription elongation factor that is known to rescue backtracked complexes[37–39]. GreB can stimulate the intrinsic cleavage activities of RNAP, leading to the removal of the 3′-end of the RNA and

alignment of the newly generated RNA 3′-end with the catalytic site, reactivating transcription. We thus conducted transcription assays in the presence of 1 μM GreB. When RNAP encountered dCas from the PAM-distal side, the transcription read-through efficiency increased substantially, from 43% to 70% for dCas9 and from 47% to 73% for dCas12a. It is interesting that, when RNAP encountered dCas from the PAM-proximal side, the read-through efficiency remained essentially zero for both dCas9 and dCas12a. Our bulk transcription assays show a similar effect of GreB on the polarity of transcription read-through (Extended Data Fig. 6).

This shows that backtracking was probably the main cause of RNAP stalling at a dCas roadblock from the PAM-distal side. Although transcription through a bound dCas from the PAM-distal side is facilitated by GreB, transcription through dCas from the PAM-proximal side encounters an almost insurmountable obstacle and cannot be rescued by GreB. Thus, in the presence of GreB, a bound dCas becomes an even more highly asymmetrical and polar barrier to transcription. This ultimately results from a bound dCas complex having an unprotected DNA bubble that can be rezipped and collapsed by RNAP.

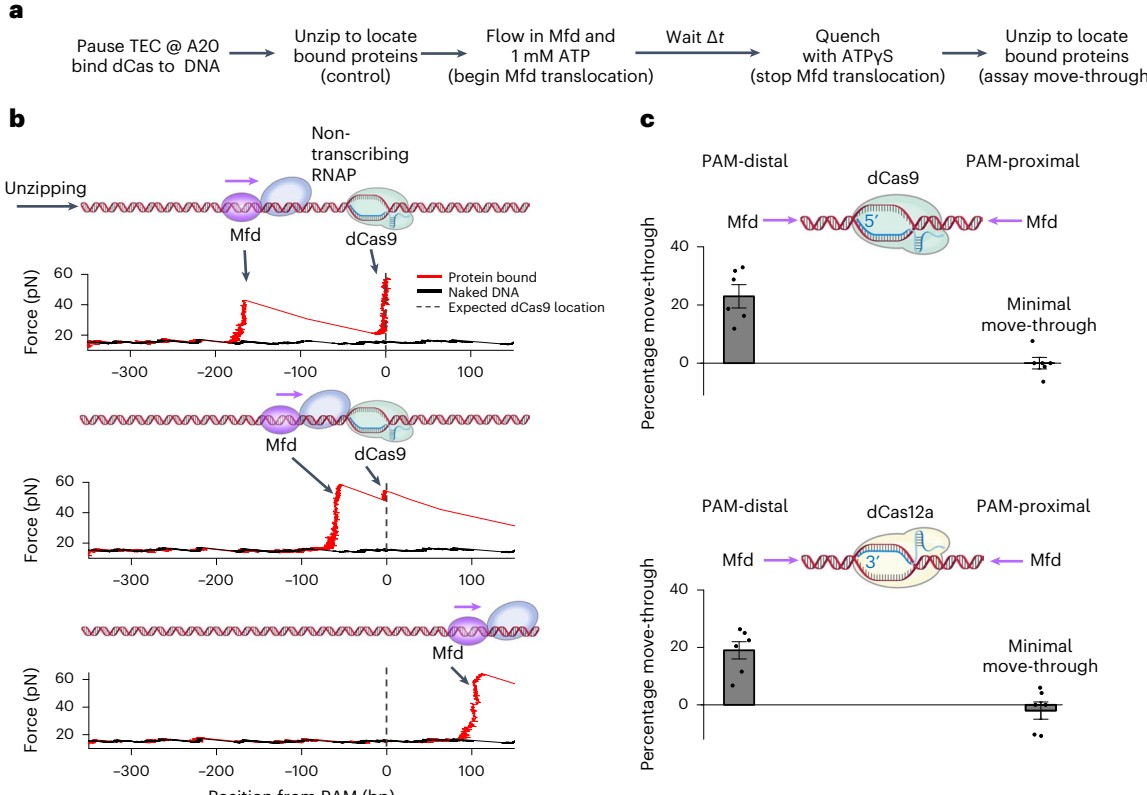

**Fig. 3 | Mfd moving through a bound dCas complex. a**, Flowchart of a single-molecule Mfd translocation assay. Some DNA tethers were used as controls to assay bound protein locations before ATP addition. Other tethers were used to assay bound protein locations after an ATP chase time of $\Delta t$ = 8 min. During the ATP chase, Mfd was able to translocate along DNA, probably still interacting with a nontranscribing RNAP. DNA was always unzipped in the same direction as Mfd translocation. **b**, Representative traces of Mfd colliding with dCas9 and moving past dCas9, when approaching dCas9 from the PAM-distal side. Naked DNA traces are shown in black. **c**, Mfd move-through efficiency for Mfd

encountering a bound dCas from either the PAM-distal side or the PAM-proximal side. Results from both dCas9 (top) and dCas12a (bottom) are shown. For each sample chamber, both control traces and noncontrol traces were taken to obtain the move-through efficiency for that chamber. Each type of experiment was repeated using $n$ = 6 biologically independent sample chambers. Move-through values were calculated for each sample chamber (black dots), and the mean value and s.e.m. of these repeats are also shown. Source data containing traces for **b** and Mfd move-through values for **c** are provided.

## A DNA translocase exhibits the same polarity

An important prediction of the hypothesized mechanism is that a bound dCas should be a polar barrier not just to RNAP, but to any DNA translocase capable of rezipping downstream DNA. To test this possibility, we required a translocase to approach a bound dCas from a defined direction. *E. coli* Mfd met this requirement because it interacts with a TEC stalled at a defined location, making it possible to control the position and orientation of translocation[26,40–42]. In the presence of ATP, Mfd can bind to the stalled TEC and forward translocate to disrupt the TEC, before processively continuing translocation in the same direction as the disrupted TEC.

Using this method of loading Mfd on to DNA, we found that, in ~85% of traces that initially contained a TEC, Mfd remained associated with DNA and translocated processively along DNA over a long distance at a rate of 2.2 bp s$^{-1}$ (Extended Data Fig. 7). The nontranscribing RNAP was presumably associated with Mfd[26,43–45], although its conformation remains undetermined. This control experiment demonstrates that Mfd can serve as a translocase and approach a bound dCas with the start and end of the translocation under the control of the ATP chase and quench.

To examine whether Mfd experiences a bound dCas as a polar barrier, we performed experiments similar to those presented in Fig. 2, except with an active Mfd instead of RNAP (Fig. 3a). As Mfd translocation is substantially slower than RNAP translocation (compare Extended Data Figs. 2b and 7b), Mfd was allowed to translocate for

480 s, so that most Mfd should reach a bound dCas before the reaction was quenched. The outcomes of Mfd collision with dCas9 were then assayed using the unzipping mapper.

Figure 3b shows example traces of Mfd approaching dCas9 from the PAM-distal side. One example trace shows a force peak detected before dCas9, consistent with Mfd not having reached the bound dCas. A second example trace shows a force peak detected immediately before a bound dCas9, consistent with Mfd colliding with dCas9. A third example trace shows a bound protein detected downstream of the dCas9-binding site, consistent with Mfd-mediated removal of dCas9 and continued translocation.

We classified the traces into different categories to determine Mfd move-through efficiency when Mfd encountered dCas9 or dCas12a from either the PAM-distal side or the PAM-proximal side (Fig. 3c and Extended Data Fig. 5). For either dCas, Mfd move-through efficiency was ~20% when encountering the bound dCas from the PAM-distal side and undetectable when encountering dCas from the PAM-proximal side. Thus, Mfd senses the same polarity as RNAP, providing strong evidence for the hypothesized mechanism of the dCas roadblock polarity.

We also noted that, when encountering a dCas from the PAM-distal side, Mfd showed a lower move-through efficiency than RNAP. We attribute this to the difference in the stability of two motor proteins when working against a strong roadblock. Although RNAP can remain stably bound to the substrate, thus allowing for multiple attempts to overcome the barrier, Mfd may tend to dissociate when working

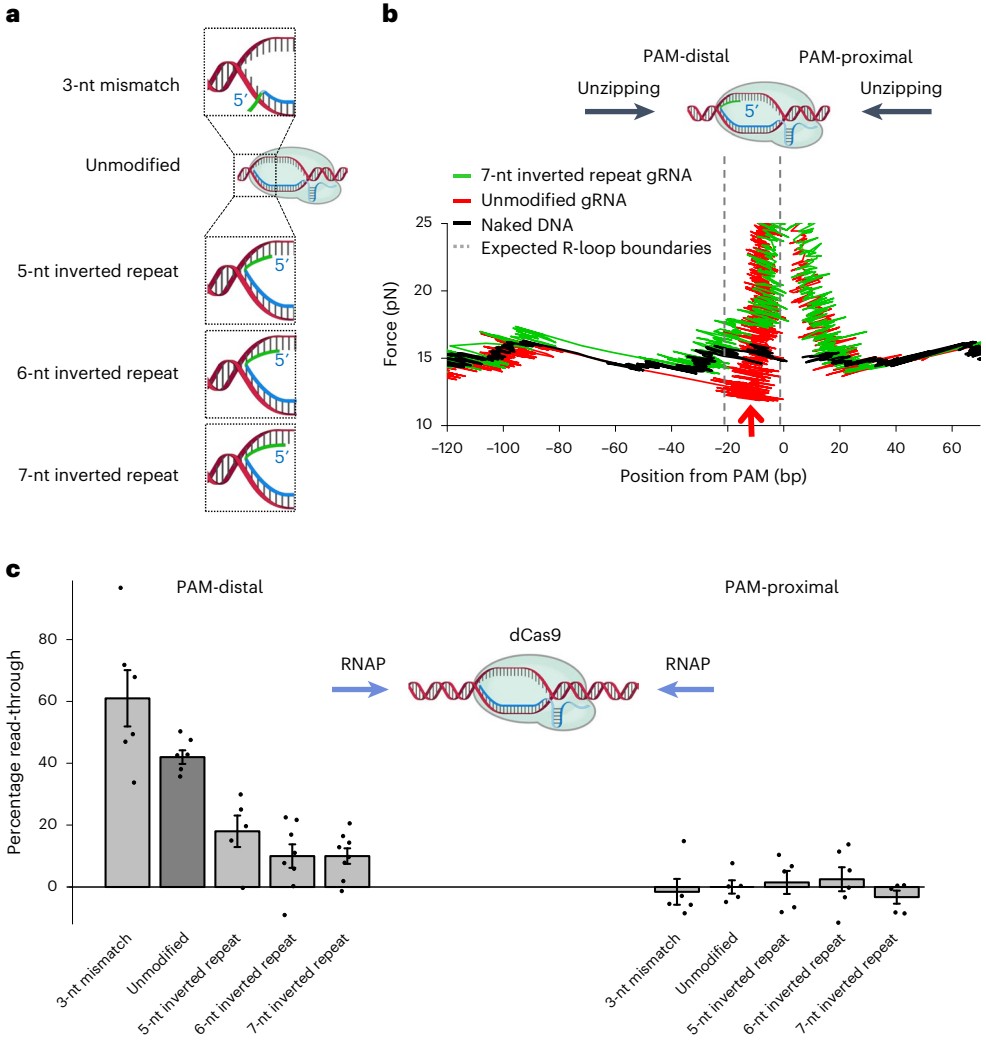

**Fig. 4 | Modulation of transcription read-through of a bound dCas complex via gRNA modifications. a**, Cartoons depicting the four types of modified gRNA. **b**, Representative unzipping mapper traces that highlight the force signature difference between a bound dCas9 containing a modified gRNA with a 7-nt inverted repeat (IR) and a bound dCas9 containing an unmodified gRNA. Vertical dashed lines bracket the position of the gRNA-DNA hybrid. Naked DNA traces are shown in black. The red arrow indicates the location of the unzipping force dropping below the naked DNA baseline for the trace with an unmodified gRNA. **c**, Transcription read-through efficiency for RNAP encountering a bound dCas from either the PAM-distal side or the PAM-proximal side. DNA was always unzipped in the same direction as RNAP translocation. For each sample chamber, both control traces and noncontrol traces were taken to obtain the read-through efficiency for that chamber. Each type of experiment was repeated using $n$ biologically independent sample chambers: dCas9 PAM-distal, $n = 6$ (3-nt mismatch), $n = 6$ (unmodified), $n = 5$ (5-nt IR), $n = 8$ (6-nt IR) and $n = 8$ (7-nt IR); dCas9 PAM-distal, $n = 5$ (3-nt mismatch), $n = 5$ (unmodified), $n = 5$ (5-nt IR), $n = 6$ (6-nt IR) and $n = 5$ (7-nt IR). Read-through values were calculated for each sample chamber (black dots) and the mean value and s.e.m. of these repeats are also shown. Source data containing traces for **b** and transcription read-through values for **c** are provided.

against a strong roadblock[26], reducing its opportunity to continue to work against the barrier.

## Modulation of transcription roadblock read-through

Our data also suggest strategies to modulate dCas roadblock polarity to transcription. For example, transcription read-through from the PAM-distal side of a dCas complex relies on disruption of the R-loop and collapse of the DNA bubble, which critically depend on gRNA interactions with DNA. Thus, if a gRNA can be modified to increase or decrease the stability of the R-loop, then transcription read-through may be downregulated or upregulated.

To increase the stability of the R-loop of a bound dCas9, we extended the 5′-end of the original gRNA with an inverted repeat sequence (Fig. 4a). This modified gRNA could form an extended R-loop that straddles across the DNA bubble's two strands. For RNAP

to transcribe through dCas9 complex containing this modified gRNA, RNAP must disrupt both the RNA-DNA hybrid on the template strand and the RNA-DNA hybrid on the nontemplate strand. The reinforced resistance by the two RNA-DNA hybrids should make it more difficult for RNAP to rewind and collapse the DNA bubble of the dCas9 complex.

We examined how such a modified gRNA impacted dCas9 binding to DNA by unzipping through the bound dCas9 using the unzipping mapper. Figure 4b shows one set of example traces of a bound dCas9 containing a gRNA with a 7-nt inverted repeat sequence at the 5′-end. Although this modified gRNA resulted in little change in the force signature for unzipping from the PAM-proximal side, the drop in force that was observed with the original gRNA for unzipping from the PAM-distal side was no longer present and the unzipping force at the expected hybrid location was slightly above that of the naked DNA baseline. This observation provides evidence for the presence of an

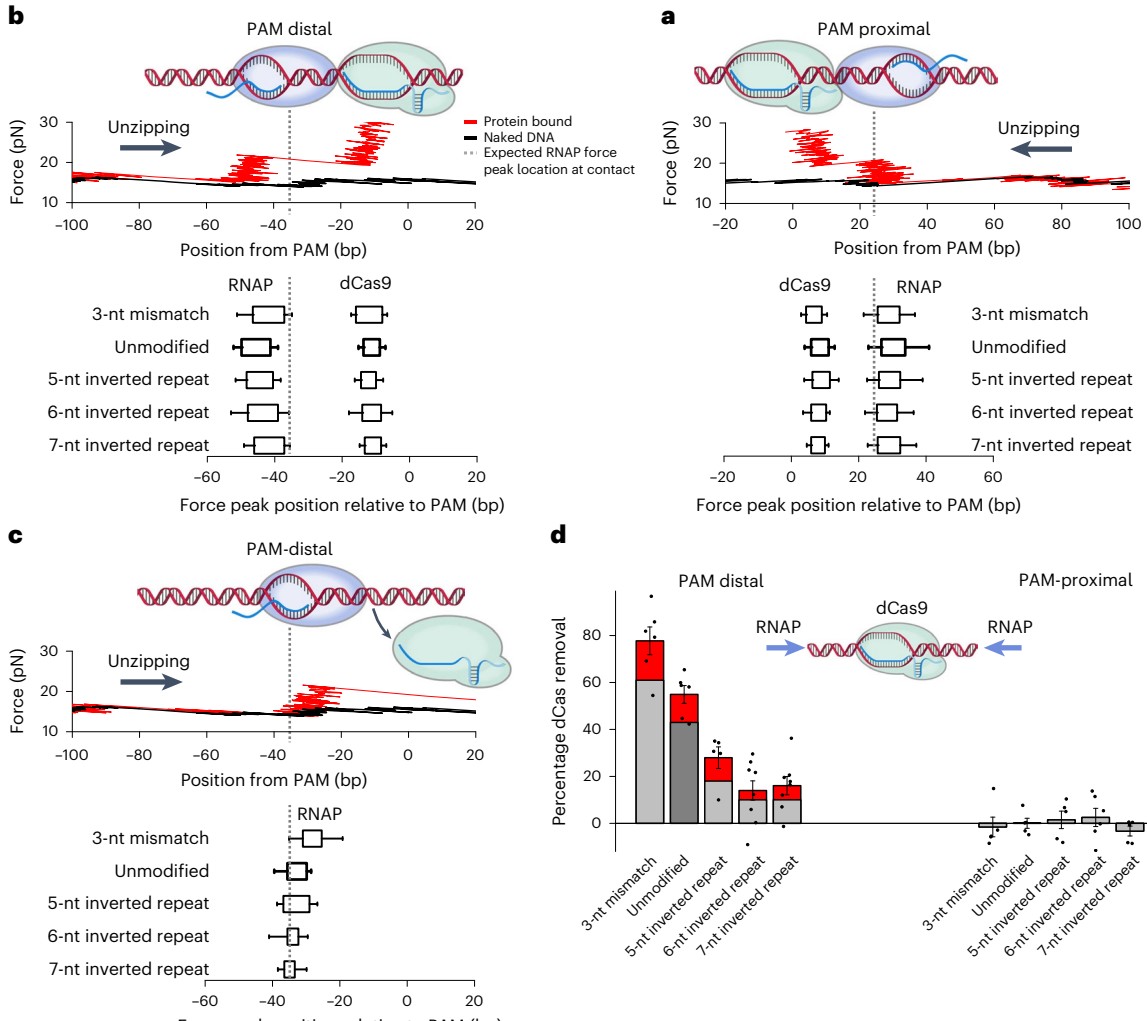

**Fig. 5 | The removal of dCas mediated by RNAP invasion of the dCas R-loop.** **a**, RNAP collision with dCas9 from the PAM-proximal side without dCas9 removal. Top, an example unzipping trace. Bottom, force peak locations of RNAP and dCas9. The dashed lines indicate the expected RNAP force peak position when RNAP contacts dCas9 (Extended Data Fig. 1a). Each box plot represents the 25th–75th percentiles of force peak positions of $n$ biologically independent traces with error bars indicating s.d.: $n = 88$ (3-nt mismatch), $n = 114$ (unmodified), $n = 103$ (5-nt inverted repeat (IR)), $n = 108$ (6-nt IR) and $n = 103$ (7-nt IR). **b**, RNAP collision with dCas9 from the PAM-distal side without dCas9 removal. Top, an example unzipping trace showing the stalled RNAP and dCas9 force peaks. Bottom, force peak locations of RNAP after collision with dCas9. Each box shows the 25th–75th percentiles of force peak position distribution with error bars indicating s.d.: $n = 14$ (3-nt mismatch), $n = 77$ (unmodified), $n = 83$ (5-nt IR), $n = 157$ (6-nt IR) and

$n = 163$ (7-nt IR). **c**, RNAP collision with dCas9 from the PAM-distal side with dCas9 being removed. Top, an example unzipping trace showing the stalled RNAP force peak with the dCas9 force peak being absent. Bottom, force peak locations of RNAP and dCas9. Each box shows the 25th–75th percentiles of force peak position distribution with error bars indicating s.d.: $n = 23$ (3-nt mismatch), $n = 22$ (unmodified), $n = 12$ (5-nt IR), $n = 15$ (6-nt IR) and $n = 12$ (7-nt IR). **d**, Efficiency of dCas9 removal with different gRNA modifications. The percentage with removal due to transcription read-through (gray, from Fig. 4c) and the percentage with removal but without read-through (red) are stacked. The percentage dCas removal values were calculated for each sample chamber (black dots), and the mean value and s.e.m. of these repeats are also shown. The same data as in Fig. 4c were used for this analysis, so the sample statistics are identical to those for Fig. 4c. Source data for **a**–**d** are provided.

R-loop that bridges the two DNA strands due to the presence of modified gRNA complexed with the bound dCas9. For the unzipping fork to proceed, the RNA–DNA hybrids formed at both DNA strands must be disrupted, elevating the force required for unzipping. Although this gRNA sequence could also form a short RNA hairpin at the 5′-end, about 70% of the unzipping traces did not show a drop in force when unzipped from the PAM-distal side, as in Fig. 4b, suggesting that the RNA configuration that straddles the DNA fork may be more stable than that with an RNA hairpin.

We have also determined how dCas9 containing such a modified gRNA modulates transcription read-through by repeating the assays outlined in Fig. 2 using extended gRNAs containing a 5-nt, 6-nt or 7-nt inverted repeat (Fig. 4c and Extended Data Figs. 2e and 8). For all three

modified gRNAs, transcription read-through from the PAM-proximal side remained essentially zero, whereas transcription read-through from the PAM-distal side decreased from 43% to 18%, 10% and 10% for the 5-nt, 6-nt and 7-nt inverted repeat gRNA, respectively. To test whether the observed PAM-distal barrier enhancement was merely a result of the extension of the gRNA, we conducted control experiments using a gRNA with an extension that is not complementary to the unmodified gRNA, so that the extended sequence cannot hybridize with the DNA bubble of a bound dCas9 or DNA in the vicinity. It is interesting that we detected a notable increase in read-through from the PAM-distal side without any detectable change in read-through from the PAM-proximal side, suggesting that the extended noncomplementary sequence hangs away from the R-loop of a bound dCas9

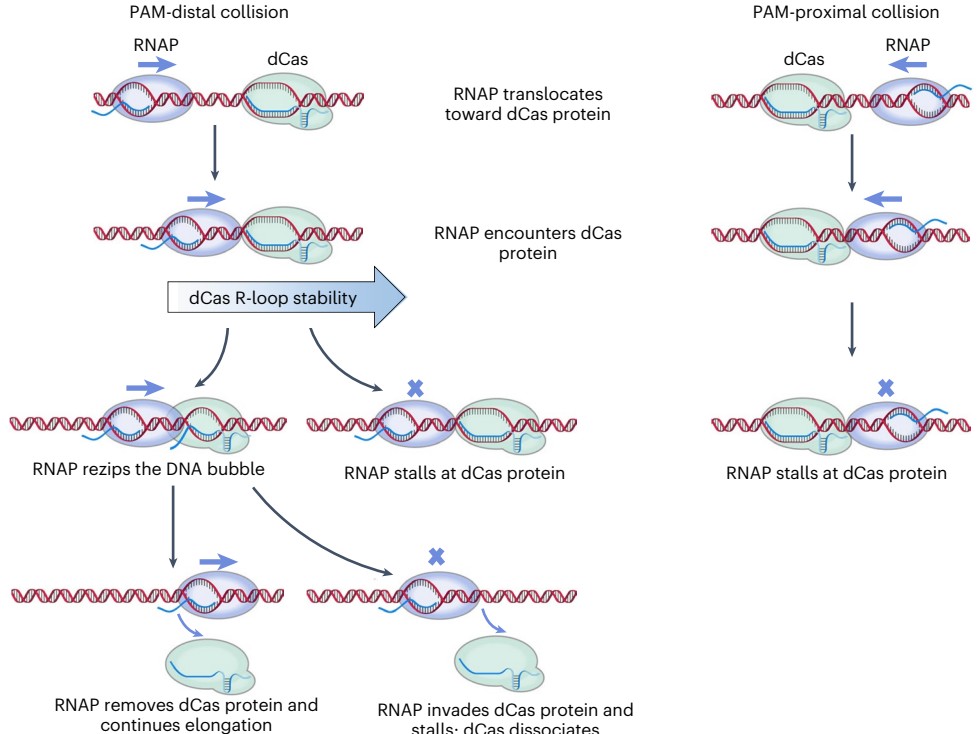

**Fig. 6 | Mechanism of CRISPR roadblock polarity to transcription.** When RNAP encounters a bound dCas from the PAM-distal side, RNAP may rezip the DNA bubble of the bound dCas, leading to R-loop disruption and DNA bubble collapse. The subsequent dCas removal allows transcription read-through. In contrast, when RNAP encounters a bound dCas from the PAM-proximal side, the DNA bubble of the bound dCas is not directly accessible to RNAP, making the dCas a strong barrier to transcription. Note that this mechanism is illustrated using a dCas9 cartoon, but the same mechanism also applies to dCas12a.

and facilitates the start of RNA-DNA separation during RNAP invasion (Extended Data Fig. 9). Thus, the barrier enhancement from a modified gRNA with an inverted repeat should not be a result of the mere extension of the gRNA.

To determine whether transcription read-through from the PAM-distal side of a bound dCas9 can also be upregulated, we introduced a 3-nt mismatch to the gRNA at its 5′-end (Fig. 4c and Extended Data Fig. 8). This mismatch should weaken the RNA-DNA hybrid, making it easier for RNAP to rezip the DNA in the bubble by disrupting the RNA-DNA hybrid. Indeed, we found that transcription read-through from the PAM-distal side increased from 43% to 61% with the 3-nt mismatch gRNA. The increase in the read-through efficiency also reflects a weakened binding of dCas9. Consistent with this, we found that a bound dCas9 containing a 3-nt mismatch showed a much faster spontaneous dissociation rate than a bound dCas9 containing an unmodified gRNA (Supplementary Table 2 and Extended Data Fig. 2e).

Collectively, these results clearly show that transcription read-through from the PAM-distal side of dCas9 can be considerably impacted via gRNA modifications. This finding also serves as strong evidence for R-loop disruption and DNA bubble collapse as a mechanism of transcription read-through.

## Modulation of transcription roadblock removal
Thus far, we have characterized the polarity of the dCas roadblock to transcription read-though, which requires the removal of the roadblock by RNAP, followed by transcription through the dCas-binding site. An alternative characterization of the roadblock polarity is the efficiency of transcription roadblock removal, which requires the removal of the roadblock by RNAP but does not require RNAP to read through the dCas-binding site. Roadblock removal may be more relevant to applications that require knowledge of whether the

roadblock is still bound, because removal of the roadblock could signal more efficient accessibility by subsequent processes to the blocked region.

Roadblock removal includes transcription read-through and an additional scenario where RNAP collided with and removed the dCas, but then stalled. To examine this, we focused on transcription data with an RNAP force signature near the expected dCas9-binding site, corresponding to stalled RNAP after collision with dCas9 (Fig. 5). Note that these traces did not result in read-through.

For PAM-proximal collisions, all traces showed both a bound RNAP and a dCas9 (Fig. 5a), indicating that, although RNAP collided with dCas9, RNAP could not remove it. Traces in this category showed a force signature consistent with the footprint of RNAP having no overlap with that of dCas9 (Extended Data Fig. 1a). The spread in the RNAP position may be a result of RNAP backtracking after collision with dCas9.

For PAM-distal collisions, the traces fall into two distinct categories. Just as with the PAM-proximal collisions, one category of traces shows both a bound RNAP and a dCas9 (Fig. 5b), consistent with the footprint of RNAP having no overlap with that of dCas9. However, the other category of traces shows only a bound RNAP, indicating that, after RNAP collided with dCas9, RNAP removed dCas9, but was then stalled in the process (Fig. 5c). In these traces, the footprint of RNAP showed a clear overlap with the expected dCas9 footprint, indicating substantial invasion of RNAP into the dCas complex, which was subsequently dissociated. It is interesting that the distance of this invasion decreases with an increase in the stability of the R-loop of dCas9. For example, for a dCas9 complex with a 3-nt mismatched gRNA, the invasion was about 7 nt into the R-loop of dCas9, whereas for a dCas9 complex with a 7-nt inverted repeat gRNA, there is minimal invasion. These data show that, the weaker the R-loop of dCas9, the easier it is for RNAP to invade and rezip the DNA bubble of dCas9, ultimately removing dCas9. Consistent

with this, the fraction of collision traces with dCas9 removed also decreased with an increase in the R-loop stability of dCas9 (Fig. 5d).

The overall roadblock removal efficiency, considering both the collision traces and the read-through traces, shows that dCas9 removal is also polar (Fig. 5d). In comparison with the read-through efficiency (Fig. 4c), the roadblock removal efficiency shows an even greater polarity and this polarity can also be modulated via modification of the gRNA (Fig. 5d) and transcription factors, such as GreB (Extended Data Fig. 10).

## Discussion

To fully realize the potential of CRISPR technology, it is crucial to obtain an in-depth mechanistic understanding of Cas-binding stability. Using the CRISPRi system, this work presents high-resolution structural features of dCas-DNA interactions, elucidates the nature of dCas removal by motor proteins and details the highly tunable nature of dCas removal through modifications of the gRNA.

We discovered a mechanistic explanation for the roadblock polarity that dCas presents to transcription in CRISPRi (Fig. 6). When approaching a bound dCas from the PAM-distal site, RNAP may be able to remove the dCas by disrupting the R-loop, rezipping the DNA bubble and removing the dCas. In contrast, when approaching a bound dCas from the PAM-proximal site, the R-loop is inaccessible to RNAP and thus RNAP seems to encounter an insurmountable obstacle. We show that this explanation for the polarity holds for both dCas9 and dCas12a, which have their target sequences of gRNA located at the 5′-end and the 3′-end, respectively. We also predicted that other dsDNA translocases should sense the same roadblock polarity as RNAP and verified this prediction using Mfd. We further demonstrate that both transcription read-through and roadblock removal by transcription from the PAM-distal side can be modulated by gRNA modifications that alter the R-loop stability of a bound dCas9 complex.

We also show that GreB can facilitate RNAP read-through when RNAP encounters a bound dCas from the PAM-distal side, but has no detectable effect on read-through when RNAP encounters a bound dCas from the PAM-proximal side, demonstrating that dCas is a highly asymmetrical and polar barrier to transcription. In vivo, other transcription factors may interact with RNAP to either up- or downregulate transcription read-through.

In addition to CRISPRi, dCas complexes have also been used to hinder replication. In contrast to transcription, this hindrance was not found to be polar[9,10]. Our proposed mechanism allows for both the presence of polarity for transcription and the absence of polarity for replication. For RNAP to read through a dCas roadblock from the PAM-distal side, RNAP must rezip the DNA downstream to collapse the R-loop of the dCas complex, and thus the ability to rezip is crucial for read-through. In contrast, a replisome relies on its helicase to unzip DNA to separate strands and, therefore, cannot rezip to collapse the R-loop of a bound dCas complex. To our knowledge, this is the first mechanistic explanation of these apparently disparate findings of dCas roadblock polarity for transcription and replication.

Beyond CRISPRi, dCas proteins are used in a host of other cellular applications. For example, they can be fused to other proteins to direct them to specific loci. In those applications, understanding stability and tuning the dwell time of dCas are integral to the sensitivity and efficiency of the assays[46,47]. Our findings suggest that inverted repeat modifications of gRNA sequences may increase the overall stability of dCas9 and improve this technology.

Besides engineered dCas proteins, naturally occurring Cas proteins without any inherent nuclease activity are known to direct DNA transposition. In these transposon-associated CRISPR–Cas systems, Cas binding is followed by recruitment of multiple other enzymes that then direct transposition. These systems have been repurposed for gene editing[48–50]. The stability of bound Cas complexes in these systems should be governed by the same mechanism described in the present study and modulation of this stability could optimize the efficiency of transposition and gene editing.

Although our work focuses on dCas proteins, these findings may also have broader implications for gene editing. For example, when insertions/deletions are created via nonhomologous end-joining (NHEJ), gene editing may be enhanced by removal of post-cleavage Cas9 via transcription machinery, which exposes a double-strand break for repair by NHEJ[51]. However, this removal may not be desirable if the goal is to utilize homology-directed repair (HDR) to perform precise edits. Cas9 removal may contribute to the observed high probability of the NHEJ pathway selected over the HDR pathway[51,52]. Cas nuclease removal can also probably be modulated using the same strategy of gRNA modifications as we demonstrated here. Successful modulation of Cas9 removal efficiency may offer better control over the partition between the HDR and NHEJ pathways.

The present study represents a mechanistic explanation of dCas roadblock polarity and demonstrates the importance of R-loop stability. Our work suggests two avenues that impact Cas binding: stability of the R-loop and access to the R-loop. Strategies for optimizing and customizing Cas binding may include modifications to the gRNA to alter the gRNA-DNA interactions and modulation of protein-DNA interactions to regulate R-loop accessibility. Understanding Cas-binding stability also provides a framework to impact the efficiencies of CRISPR applications.

## Online content

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

## Methods

### Protein purification

*E. coli* RNAP was purified using tagged purification[26,53,54]. Briefly, RNAP was expressed at low levels in 5α-competent *E. coli* (Invitrogen, catalog no. 18265-017) transformed with the plasmid pKA1 in Superbroth (25 g l⁻¹ of tryptone (Sigma-Aldrich, catalog no. T2559), 15 g l⁻¹ of yeast extract (Sigma-Aldrich, catalog no. Y1626), 5 g l⁻¹ of NaCl (Sigma-Aldrich, catalog no. S3014) with 100 µg ml⁻¹ of ampicillin (Sigma-Aldrich, catalog no. A0166)) for 4 h until the absorption at 600 nm ($OD_{600}$) reached 2.1. Cells were induced with isopropyl β-D-1-thiogalactopyranoside (IPTG; RPI, catalog no. I56000-50) to a final concentration of 1 mM for 4 h. Cells were lysed and sonicated on ice in small aliquots (<20 ml). with a macro-tip on a Branson Sonifier 250 and 60% duty cycle. Sonicated cells were centrifuged to pellet cell debris and the pellet was discarded. To precipitate nucleic acids and their bound proteins out of solution, cleared 5% (w/v) polyethyleneimine (PEI), pH 7.9 (made from 50% stock; Sigma, catalog no. P3143) was slowly added to the supernatant to a final concentration of 0.4% (w/v). The DNA with bound RNAP was pelleted from the solution, washed 5× with a buffer containing 350 mM NaCl (J.T.Baker, catalog no. 4058-01) and RNAP was eluted from the PEI and DNA with a buffer containing 1 M NaCl. The eluted RNAP was purified to homogeneity with chromatography using three columns: a HiPrep Heparin FF 16/10 column (GE Healthcare, catalog no. 28-9365-49), a HiPrep 26/60 Sephacryl S-300 HR column (GE Healthcare, catalog no. 17-1196-01) and a QIAGEN Ni-NTA Superflow column (catalog no. 30410). Fractions that contained holo-RNAP were pooled, concentrated and dialyzed into RNAP storage buffer (50 mM Tris-HCl, pH 8.0 (J.T.Baker, catalog nos. 4103-01 and 4109-01), 100 mM NaCl, 1 mM EDTA (Invitrogen, catalog no. 15508-013) 50% (v/v) glycerol (J.T.Baker, catalog no. 4043-00) and 1 mM dithiothreitol (DTT; Invitrogen, catalog no. 15508-013)) and ultimately stored at −20 °C.

*E. coli* GreB was purified using tagged purification[54]. Briefly, Plasmid pES3, encoding GreB-6xHis in pET-28b(+) (4), was transformed into BL21(DE3) (Invitrogen, catalog no. 44-0049) cells for protein overexpression. Cells were grown at 37 °C in Luria broth (Affymetrix, catalog no. 75854) with 50 µg ml⁻¹ of added kanamycin (Sigma, catalog no. K0254) at 37 °C until the $OD_{600}$ was between 0.6 and 0.8; induction was then carried out with 1 mM IPTG (Roche, catalog no. 10724815001). After 3 h at 37 °C, cells were harvested by centrifugation and stored at −80 °C. To purify GreB, cells were thawed on ice and resuspended in GreB lysis buffer (50 mM Tris-HCl (Thermo Fisher Scientific, catalog nos. BP154 and BP153), pH 6.9, 500 mM NaCl (Thermo Fisher Scientific, catalog no. BP358) and 5% v/v glycerol (Thermo Fisher Scientific, catalog no. BP229)), using lysozyme (300 µg ml⁻¹; Sigma-Aldrich, catalog no. 10837059001) and EDTA-free protease inhibitor cocktail (Roche, catalog no. 11873580001). The cells were placed on ice for 1 h and then briefly sonicated for more complete lysis. The extract was centrifuged (24,000*g* for 20 min at 4 °C) and twice passed through a 0.45-µm filter. An Ni-NTA agarose (Invitrogen, catalog no. R90115) column was used for GreB isolation and GreB lysis buffer with 200 mM imidazole was used for elution. The eluate was then run on a Superdex 200 column (Cytivia, catalog no. 28990944) with elution buffer (10 mM Tris-HCl, pH 8.0, 500 mM NaCl, 1 mM DTT, 1 mM EDTA and 5% v/v glycerol). Dialysis was performed into GreB storage buffer (10 mM Tris-HCl, pH 8.0, 200 mM NaCl, 1 mM DTT, 1 mM EDTA and 50% v/v glycerol) and stored at −80 °C after a flash-freeze in liquid nitrogen.

*E. coli* Mfd was purified using tagged purification[55]. Briefly, a pET plasmid was used to overexpress *Eco* Mfd with its amino terminus His₆ tagged. This plasmid was transformed via heat shock at 42 °C for 40 s into Rosetta(DE3) pLysS cells (Novagen, catalog no. 70956-M); 1 mM IPTG (Goldbio, catalog no. I2481C) was added to cells ($OD_{600}$ = 0.67) for 4 h at 30 °C to induce protein expression. For harvesting, cells were centrifuged and pellets were resuspended in a lysis buffer (50 mM Tris (MP, catalog no. 103133), pH 8.0, 500 mM NaCl (Thermo Fisher Scientific,

catalog no. S271-500), 15 mM imidazole (MP, catalog no. 02102033-CF), 10% (v/v) glycerol (Thermo Fisher Scientific, catalog no. BP2294), 2 mM β-mercaptoethanol (β-ME; Sigma-Aldrich, catalog no. M6250), 1 mM phenylmethylsulfonyl fluoride (Sigma-Aldrich, catalog no. P7626) and protease inhibitor cocktail (cOmplete, EDTA-free; Roche, catalog no. COEDTAF-RO)) and subsequently lysed on a French press. Lysate was flown over a Ni²⁺-charged Hitrap IMAC column (Cytiva, catalog no. 17524802) and eluted over the course of a 0- to 200-mM imidazole gradient. Post-nickel column dialysis was performed in a buffer containing 20 mM Tris, pH 8.0, 100 mM NaCl, 10% (v/v) glycerol, 5 mM EDTA (Sigma-Aldrich, catalog no. E5134) and 10 mM β-ME, and the dialyzed sample was loaded on to a Hitrap Heparin column (Cytiva, catalog no. 28-9893-35). Elution was performed over the course of a 100-mM to 2-M NaCl gradient, and the resulting sample was further purified on a HiLoad 16/600 Superdex200 size exclusion chromatography column in a buffer containing 20 mM Tris, pH 8.0, 500 mM NaCl and 10 mM DTT (Goldbio, catalog no. DTT10). Glycerol was added to the purified Mfd to reach a final concentration of 20% (v/v) and the sample was flash frozen in liquid nitrogen and finally stored at −80 °C.

### Guide RNA preparation

Cas9 single guide RNAs were custom synthesized by Sigma-Aldrich, and purified using 8% denaturing urea polyacrylamide gel electrophoresis (urea–PAGE), similar to previous descriptions[16,56]. Cas12a gRNA was prepared by cloning the Cas12a gRNA sequence[57] (Supplementary Table. 2) into a pUC19 plasmid, containing a T7 promoter and a downstream hepatitis delta virus (HDV) ribozyme sequence[58], by site-directed mutagenesis (Supplementary Table 3). T7 transcription templates were generated from the cloned plasmids via PCR with Q5 DNA polymerase (New England Biolabs (NEB), catalog no. M0491). In vitro transcription was performed for each template by incubation with T7 RNAP (NEB, catalog no. M0251) at 37 °C for 3 h, followed by incubation at 65 °C for 20 min to promote ribozyme cleavage and leave a 3′-cyclic phosphate. Products were dephosphorylated with T4 PNK (NEB, catalog no. M0201) and purified by urea–PAGE.

### Single-molecule DNA unzipping templates

Single-molecule DNA unzipping templates were generated from a pRL574 plasmid, which contains a T7A1 promoter (Supplementary Tables 3 and 4). The PAM-distal Cas9 template was identified in pRL574, 309 bp from the +20. To generate templates for the remaining three templates (Cas9 PAM-proximal, Cas12a PAM-proximal and Cas12a PAM-distal), an ~60-bp region of pRL574 was modified via site-directed mutagenesis using a protocol from NEB and Q5 DNA polymerase (Supplementary Table 3). For each template, we substituted a 63- or 64-bp DNA segment at 290 bp from the +20 for Cas9 or both Cas12a templates, respectively (Supplementary Table 4). The substituted DNA segment contained the relevant target sequence and PAM as well as 20 bp of conserved flanking DNA on either side.

Four DNA unzipping segments were amplified by PCR, digested with DraIII (NEB, catalog no. R3510), leaving a single-strand (ss)DNA overhang (TAG) and purified by 0.8% agarose gel electrophoresis (Supplementary Table 4). These templates were used as transcription templates and for PAM-distal dCas and upstream RNAP mapping. Two additional reversed unzipping segments were used for PAM-proximal dCas and downstream paused transcription complex (PTC) mapping, and generated by PCR and digested with AlwNI (NEB, catalog no. R0514; Supplementary Table 4). These DNA segments were then each ligated to a pair of dsDNA arms containing a CTA overhang at their junction[25,26]. Both DNA arms were amplified by PCR from pBR322 (NEB, catalog no. N3033) and digested by BamHI. One arm was end-labeled with biotin and the other with digoxigenin through separate Klenow reactions with biotin-14-dATP (Invitrogen, catalog no. 19524016) and digoxigenin-11-dUTP (Roche, catalog no. 11093088910), respectively. Each arm was digested with BsmBi-V2 (NEB, catalog no. R0739S), ligated

to an annealed adapter oligo and gel purified. Finally, the arms were annealed to each other at an equimolar ratio to create y-arm adapters suitable for ligation of an unzipping segment. All oligonucleotide sequences required for template creation are listed in Supplementary Table 3.

## Protein complex formation for single-molecule experiments

PTC was formed in bulk on an unzipping template which contained a promoter in the unzipping segment. The complex was paused at the A20 position via nucleotide depletion[24,26]. Briefly, a 10-nM DNA template was mixed with 50 nM RNAP in the presence of 250 µM ApU (Dharmacon, customized synthesis), 50 µM GTP (Roche, catalog no. 11140957001), ATP (Roche, catalog no. 11140965001) and CTP (Roche, catalog no. 11140922001), 1 U µl⁻¹ of Superase-in (Invitrogen, catalog no. AM2694) in transcription buffer (TB, 25 mM Tris-Cl, pH 8, 100 mM KCl, 4 mM MgCl₂ (Invitrogen, catalog no. AM9530G), 1 mM DTT, 3% glycerol (Thermo Fisher Scientific, catalog no. BP229), 150 µg ml⁻¹ of AcBSA (Invitrogen, catalog no. AM2614)). For high-resolution TEC mapping experiments, the mixture also contained 1 mM 3′-dUTP (Trilink, catalog no. N-3005)[59] which paused the complex at U21. For all experiments, the mixture was incubated at 37 °C for 30 min and then briefly placed on ice. The mixture was quickly diluted 1:100 and immediately introduced into a prepared sample chamber. To form dCas–gRNA complex, 50 nM Cas9 sgRNA or 100 nM Cas12a gRNA was denatured in RNA storage solution (Invitrogen, catalog no. AM7001) at 80 °C for 1.5 min and then placed on ice; 75 nM Sp–dCas9 (NEB, catalog no. M0652) or 300 nM As–dCas12a (IDT, off catalog) along with 1× TB was then added. The mixture was incubated at 37 °C for 10 min and then placed on ice until introduction into a prepared sample chamber. The dCas–gRNA complex was later introduced into a single-molecule sample chamber to allow dCas binding to DNA as described below.

## Single-molecule experimental procedures

For all unzipping assays, DNA tethers were formed in a sample chamber consisting of a cleaned glass coverslip as previously described[24–26]. Anti-digoxigenin (Vector Labs, catalog no. MB-7000) in TB at 16.7 µg ml⁻¹ was introduced into the chamber, allowed to incubate for 10 min at room temperature (RT) and replaced with 65 µl of TB with 10 mg ml⁻¹ of casein (Sigma-Aldrich, catalog no. C8654). After 10 min at RT, a 5-pM DNA template was introduced, allowed to incubate for 5 min at RT and later replaced with 90 µl of TB. Finally, 0.5 pM of 489-nm streptavidin-coated polystyrene beads in TB with 1 mg ml⁻¹ of casein was introduced and incubated for 10 min and RT. The buffer was replaced with 80 µl of TB. This resulted in DNA templates tethered between the surface of a coverslip via a dioxygenin and anti-dioxygenin connection and a 489-nm bead via a biotin and streptavidin connection (Fig. 1a).

The dCas–DNA complexes were formed on DNA tethers by introducing 75 µl of prepared dCas–gRNA complexes (25 pM dCas9–sgRNA or 200 pM dCas12a–sgRNA) and incubating for 10 min before replacing the chamber buffer with 90 µl of TB. For inverted repeat gRNAs, 37.5 pM dCas9–sgRNA was introduced. For 3-bp mismatched guides, 50 pM was introduced to ensure a high fraction of bound dCas9 (>90%).

For roadblock assays, PTCs and dCas–DNA complexes were formed as described using the appropriate unzipping template for the selected dCas target (Supplementary Table 2). Free dCas proteins were removed by flushing the sample chamber with 90 µl of TB. Subsequently, occupancy of each bound protein was assessed via unzipping ~40 tethers. For transcription resumption, 75 µl of TB supplemented with 1 mM NTP each (UTP; Roche, catalog no. 11140949001) and 1 mM MgCl₂ was introduced into the sample chamber. The transcription reaction was chased for 135 s before being quenched by introducing 120 µl of TB with 4 mM Mg²⁺ into the chamber. For Mfd translocase experiments, 75 µl of 166 nM Mfd with 2 mM ATP and 4 mM Mg²⁺ in TB was introduced into the sample chamber. After 480 s, the reaction was

quenched by introducing 75 µl of TB with 1 mM adenosine 5′-O-(3-thio) triphosphate (ATPγS; Sigma-Aldrich, catalog no. A1388) and 5 mM Mg²⁺. After quenching, the bound proteins were assayed by unzipping 60 or 80 tethers for transcription or translocation reactions, respectively.

## Optical trapping measurements

We used a surface-based optical trap setup[60] (Fig. 1a). For Figs. 1 and 4b, tethers were unzipped at a loading rate of 8 pN s⁻¹. For the remaining experiments, tethers were unzipped at a constant velocity of 500 nm s⁻¹. Data for all assays were acquired at 10 kHz and decimated with averaging to 1 kHz. Raw force and extension data were used to obtain the number of base-pairs unzipped via dsDNA and ssDNA elastic parameters[25,61]. The force versus number of base-pairs unzipped was then aligned to the expected unzipping theory curve to increase the accuracy and precision of locating bound protein interactions[26]. Data acquisition and conversion were performed using customized LabView 7 software and all downstream analyses were performed using customized MATLAB 7 Code.

The force peak position of a protein bound to DNA was identified as the location of a rise in vertical force that deviated from the theoretical force versus the number of base-pairs unzipped. Subsequent to transcription reactions, some force peaks near the RNAP showed a small but distinct tether-shortening event. This was attributed to the nascent transcript partially annealing to the exposed ssDNA. For traces that had this detectable shift, this slight shortening was corrected for in the location of the dCas9 peak.

All optical trapping measurements were performed in a temperature-controlled room at 23.3 °C. However, the temperature increased slightly to an estimated 25 °C owing to local laser trap heating[62]. All reactions were also carried out at an RT of 23.3 °C.

## Calculation of transcription read-through

To accurately quantify the rate of dCas read-through by a translocase in single-molecule assays, we needed to take into account the following considerations: (1) the initial state of the traces might vary slightly from sample chamber to sample chamber; (2) proteins initially bound to DNA might dissociate through a noncollision mechanism; and (3) some translocases were inactive or could not reach the collision site. Therefore, to calculate read-through rate after these considerations, we used conditional probabilities to determine how each category of traces changed after chasing.

Before chasing, we classified traces into one of four fractions. For a given sample chamber, the fraction of PTC at A20 and a dCas ($F_{A20,dCas\_i}$) was always the dominating fraction, representing typically 90% of traces, and was measured for each sample chamber. The remaining traces were categorized at PTC only, dCas only or neither, with fractions denoted as $F_{A20\_only\_i}$, $F_{dCas\_only\_i}$ and $F_{Nak\_i}$, respectively. These three minor fractions also contributed to various final observed fractions and their contributions must be accounted for (Extended Data Fig. 5).

In addition, we found that a small, but notable, fraction of the TEC and bound dCas9 dissociated during the course of the experiment in the absence of any translocase activity. We accounted for this by including the probability of TEC and dCas dissociating through a noncollision mechanism as $P_{RNAP\_diss}$ and $P_{dCas\_diss}$, respectively (Extended Data Fig. 2).

After chasing, traces were categorized into one of seven fractions. Traces that showed a TEC that had not yet reached the bound dCas were classified as either with a dCas ($F_{TEC\_up,dCas\_f}$) or without a dCas ($F_{TEC\_up\_only\_f}$). We classified a trace with a TEC that has not reached dCas as one with a detected TEC force peak >60 bp upstream from the dCas target site. Traces with a TEC < 60 bp upstream from dCas site and with a dCas present were categorized as having had a dCas–RNAP collision ($F_{Coll\_f}$). Traces with a TEC < 60 bp upstream from the dCas site but without a dCas detected were categorized as RNAP having removed dCas, but then being unable to read through ($F_{dCas\_rem\_f}$). Last, we also

categorized traces with a TEC downstream of the dCas-binding site without dCas being present ($F_{TEC\_dn\_f}$), traces with no bound proteins ($F_{Nak\_f}$) or traces that consisted of dCas only with no TEC detected ($F_{dCas\_f}$).

Due to the heterogeneities in the TEC population and bound dCas, not all TEC complexes would encounter a bound dCas. We refer to the probability that an RNAP initially escaped the A20 translocated toward, and reached a bound dCas, as the probability of being collision competent, $P_{Coll\_comp}$. We can determine this probability from the probability of TEC being collision incompetent, that is, the probability that a TEC was present at A20 or upstream of the dCas, given that RNAP did not dissociate through a noncollision mechanism. $P_{Coll\_comp}$ can be calculated as:

$$P_{Coll\_comp} = 1 - \frac{F_{TEC\_up,dCas\_f} + F_{TEC\_up\_only\_f}}{(F_{A20,dCas\_i} + F_{A20\_only\_i})(1 - P_{RNAP\_diss})}.$$

To determine the probability that a TEC was able to read through a dCas, given that the TEC was collision competent and neither protein dissociated due to a noncollision mechanism, we start with the post-chase naked DNA ($F_{Nak\_f}$) and RNAP downstream ($F_{TEC\_dn\_f}$) traces, and then take into account other pathways that also contributed to those two final observations. This gives the following equation for $P_{Read-through}$:

$$P_{Read-through} = [F_{Nak\_f} + F_{TEC\_dn\_f} - F_{Nak\_i}$$
$$-F_{A20,dCas\_i}\left(P_{Coll\_comp}(1 - P_{RNAP\_diss}) + P_{RNAP\_diss}\right)P_{dCas\_diss}$$
$$-F_{A20\_only\_i}\left(P_{Coll\_comp}(1 - P_{RNAP\_diss}) + P_{RNAP\_diss}\right)$$
$$-F_{dCas\_only\_i}P_{dCas\_diss}$$
$$/\left[F_{A20,dCas\_i}P_{Coll\_comp}(1 - P_{RNAP\_diss})(1 - P_{dCas9\_diss})\right]$$

Similarly we can find the probability that a TEC will remove the dCas from the DNA by also including the fraction of traces where TEC was found to have removed dCas but was not able to read through ($F_{dCas\,rem\_f}$). This results in the following equation for $P_{Removal}$:

$$P_{Removal} = [F_{Nak\_f} + F_{TEC\_dn\_f} + F_{dCas\,rem\_f}$$
$$-F_{Nak\_i}$$
$$-F_{A20,dCas\_i}\left(P_{Coll\_comp}(1 - P_{RNAP\_diss}) + P_{RNAP\_diss}\right)P_{dCas\_diss}$$
$$-F_{A20\_only\_i}\left(P_{Coll\_comp}(1 - P_{RNAP\_diss}) + P_{RNAP\_diss}\right)$$
$$-F_{dCas\_only\_i}P_{dCas\_diss}]$$
$$/\left[F_{A20,dCas\_i}P_{Coll\_comp}(1 - P_{RNAP\_diss})(1 - P_{dCas9\_diss})\right]$$

For the Mfd collision assays of Fig. 3, the calculation of move through is almost identical; however, we must replace $P_{RNAP\_diss}$ with the observed noncollision-related dissociation of Mfd before translocation ($P_{Mfd\_diss}$) (Extended Data Fig. 7c). We also define the cutoff for Mfd being collision competent as <70 bp because this corresponds with the approximate footprint of the complex[26]. Finally, we did not observe a population of stalled Mfd near the dCas binding after dCas removal and the removal efficiency is identical to the move-through efficiency.

### Bulk transcription assays

Bulk transcription assays were done using $^{32}$P-labeled RNA, separated by urea–PAGE[26,53,63]. Four 5′-biotinylated DNA templates, each containing the T7A1 promoter and a dCas-binding site, were amplified from the pRL574 variants using Taq Polymerase (NEB, catalog no. M0273). The resulting templates are detailed in Supplementary Table 5. The templates were bound to streptavidin-coated magnetic beads (NEB, catalog no. S1420) at a concentration of 100 nM and mixed by

rotation for 12 h at 4 °C. PTCs were made in a similar fashion, as noted above for single-molecule assays, by combining 20 nM bead-bound DNA, 100 nM RNAP, 50 µM CTP, 50 µM ATP, 30 µCi of [α-$^{32}$P]GTP (Perkin-Elmer, catalog no. BLU006H250UC), 250 µM ApU and 1 U µl$^{-1}$ of Superase-in, and incubating for 30 min at 37 °C. PTCs were then immediately washed 3× with TB. A magnetic tube rack was used to pull down PTCs and the pellet was washed and resuspended in TB. The dCas–gRNA complexes were formed similarly to single-molecule assays, added to the washed PTCs (40 nM for dCas9, 250 nM for dCas12a) and incubated at 37 °C for 10 min. The resulting PTCs and dCas complexes were then washed with TB as before to remove free dCas–gRNA. Finally, TECs primed for collision with bound dCas–gRNA were chased by adding 1 mM NTPs with or without 1 µM GreB in TB with 5 mM MgCl$_2$ for 135 s. The reaction was quenched and transcripts were released from TECs by adding 1× RNA loading dye (NEB, catalog no. B0363) and 25 mM EDTA (MP, catalog no. 194822). Magnetic beads were pulled down using a magnetic rack. The supernatant containing the transcript was removed, heated to 95 °C for 10 min and then immediately loaded on to a 20-cm 6% urea–PAGE pre-run to 55 °C using a Protean Xi Cell (BioRad). The gel was dried using a Model 583 gel dryer (BioRad), exposed to a phosphor screen (FujiFilm) for 12 h and scanned on a Typhoon 700 Imager (Cytiva). Images were linearized using ImageJ and lane profiles were analyzed using MATLAB.

### Reporting summary

Further information on research design is available in the Nature Research Reporting Summary linked to this article.

### Data availability

Relevant Source data for the main text and Extended Data Figs. are provided with this paper. All other data that support the findings of the present study are available from the corresponding author upon reasonable request.

### Code availability

All relevant code is available upon reasonable request.

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

## Acknowledgements

We thank C. Tan at Cornell University for the purified GreB, the Ailong Ke lab at Cornell University for providing us with the HDV ribozyme vector backbone to generate gRNA, the Henning Lin lab at Cornell University for use of their Typhoon Imager, and J. Roberts from Cornell University, S. Sternberg from Columbia University and the Wang Lab members for commenting on the manuscript. This work is supported by the National Institutes of Health (grant nos. R01GM136894 to M.D.W. and T32GM008267 to M.D.W.) and the National Science Foundation (grant no. MCB-1517764 to M.D.W.). M.D.W. is a Howard Hughes Medical Institute investigator.

## Author contributions

P.M.H. and M.D.W. designed the experiments. P.M.H. carried ot experiments and performed data analysis. J.T.I. assisted with data analysis. R.M.F. purified RNAP. J.B. and S.D. purified Mfd. T.T.L. helped troubleshoot the transcription experiments. G.L. assisted in early exploration of the project. M.D.W. and P.M.H. drafted the manuscript. All authors edited the manuscript. M.D.W. provided overall guidance on experimental designs and measurements.

## Competing interests

Cornell has filed for a provisional patent (no. 63/280,448) related to this manuscript with M.D.W. and P.M.H. as inventors. The other authors declare no competing interests.

## Additional information

**Extended data** is available for this paper at https://doi.org/10.1038/s41594-022-00864-x.

**Correspondence and requests for materials** should be addressed to Michelle D. Wang.

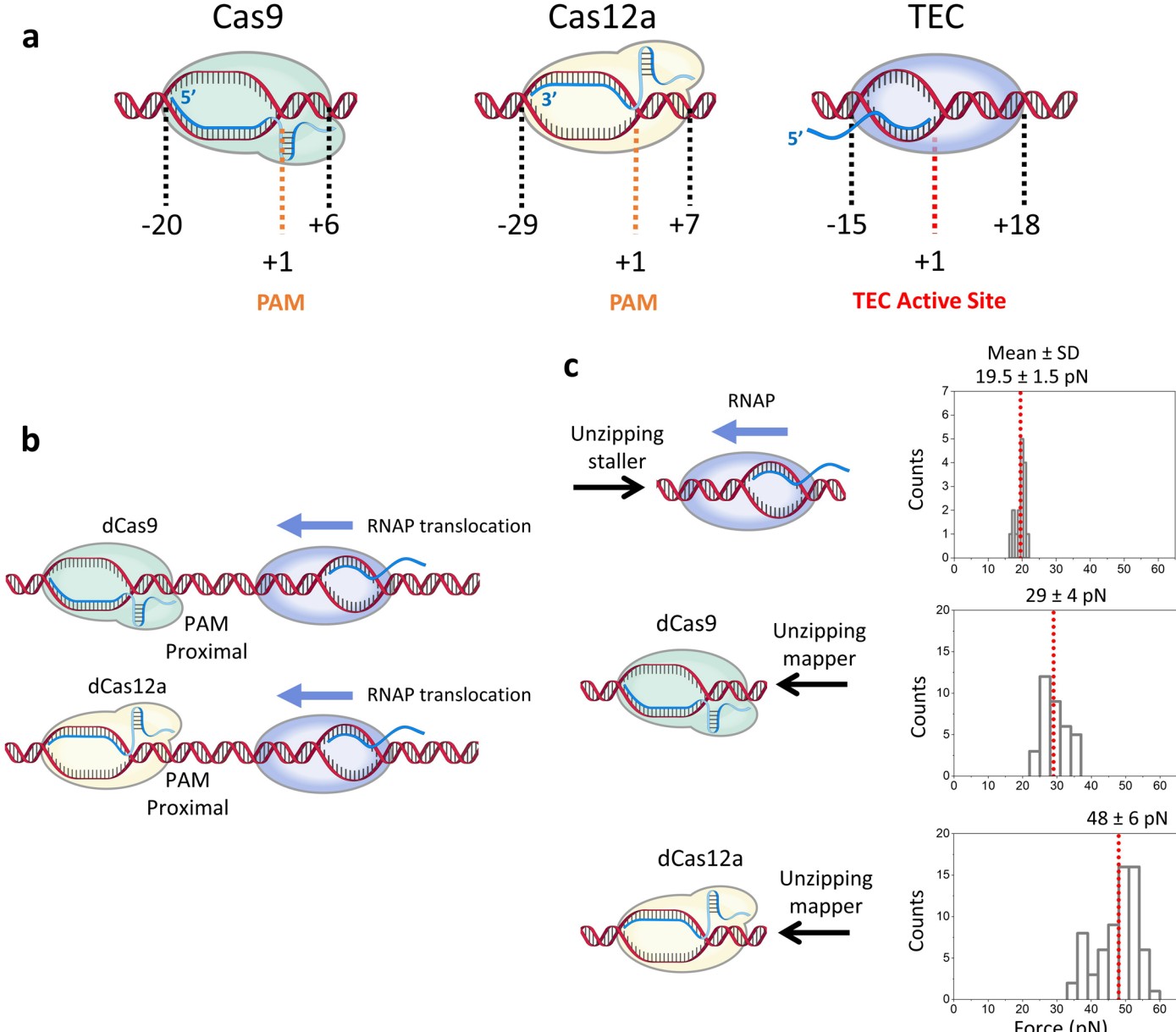

**Extended Data Fig. 1 | RNAP encountering a bound dCas protein from the PAM-proximal side. a**, Structural features of TEC and dCas. Numbers shown are our best estimates based on published structural data of dCas9[18,21,23,64], dCas12a[57,65], and TEC[29,55,66]. **b**, Cartoon of RNAP approaching the PAM proximal region of dCas9 or dCas12a. **c**, The distribution of stall forces of an actively elongating RNAP obtained using the unzipping staller method[26] is compared to the peak disruption forces from PAM-proximal unzipping of dCas proteins using the unzipping mapper technique from Fig. 1. The forces required to disrupt a bound dCas from the PAM-proximal dCas side are well above the forces that RNAP can generate working against a fork before stalling. This suggests that the dCas barrier from the PAM-proximal side is unsurpassable by RNAP. Source Data File containing histograms for c is provided.

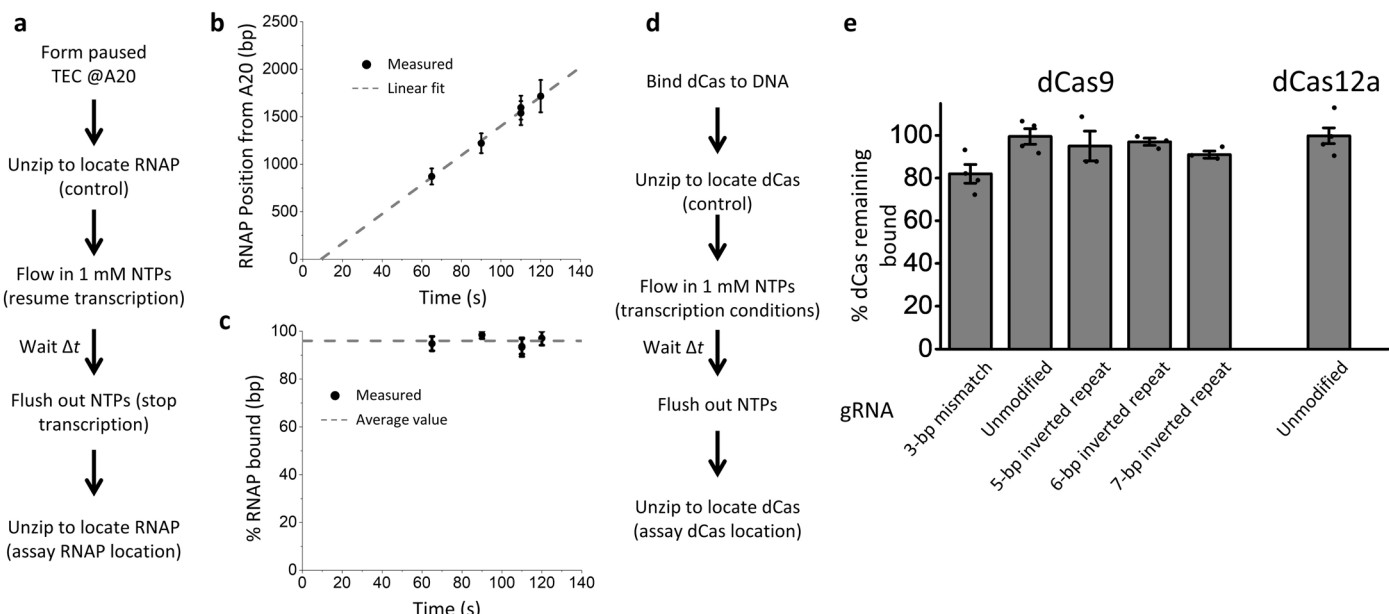

**Extended Data Fig. 2 | Transcription and dCas binding control experiments.**
**a**, Flowchart of the single-molecule assay for a control experiment to determine
RNAP speed and processivity. **b**, The mean distance RNAP traveled as a function
of chase time. $N$ number of biologically independent traces were used for analysis
at each time point: $N = 42$ at 65 s, $N = 44$ at 90 s, $N = 65$ at 110 s, and $N = 18$ at
120 s. Error bars are SEMs. The gray dashed line is a linear fit, yielding a slope of
$15.4 \pm 0.6$ bp/s for the RNAP speed. **c**, Percent of RNAP remaining on the template
normalized to initial occupancy. $N$ number of biologically independent traces
were used for analysis at each time point: $N = 58$ at 65 s, $N = 65$ at 90 s, $N = 104$ at
110 s, and $N = 34$ at 120 s. Error bars are SEMs. The average 96% occupancy (grey
dashed line) indicates that around 4% of RNAP likely dissociated upon chasing

before transcription resumption, but the remaining population remained bound
as RNAP moved down the template. **d**, Flowchart of the single-molecule assay
for a control experiment to measure the fraction of dCas remaining bound after
the introduction of transcription conditions. **e**, % of dCas remaining bound after
the quench. For each sample chamber, both control traces and non-control
traces were taken to obtain the % dCas remaining for that chamber. Each type of
experiment was repeated using $N$ biologically independent sample chambers:
Cas9, $N = 4$ (3-nt mismatch), $N = 4$ (unmodified), $N = 3$ (5-nt IR), $N = 3$ (6-nt IR),
and $N = 3$ (7-nt IR); dCas12a, $N = 4$ (unmodified). % dCas remaining values were
calculated for each sample chamber (black dots), and the mean value and SEM of
these repeats are also shown. Source Data File for b, c, and e is provided.

## dCas9

**a**

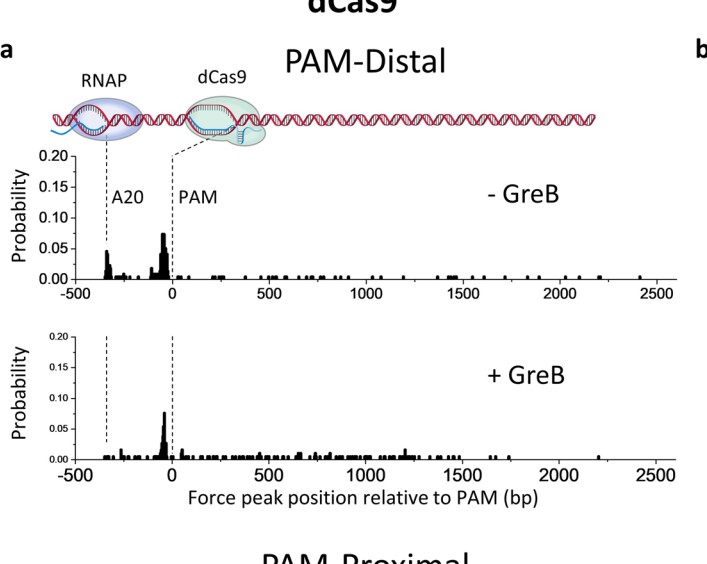

## dCas12a

**b**

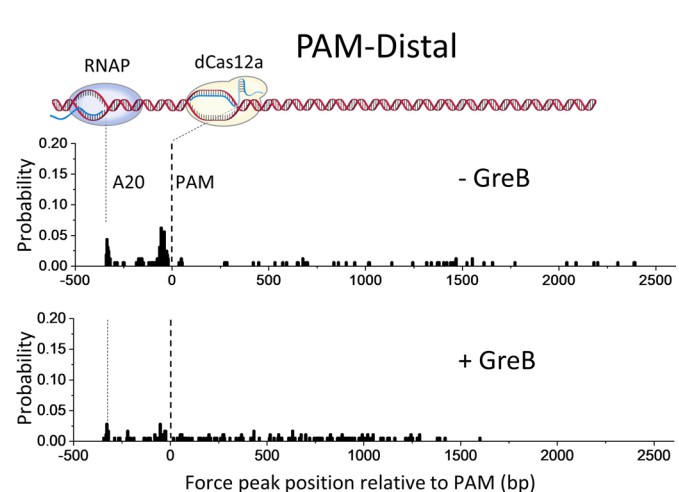

## PAM-Proximal

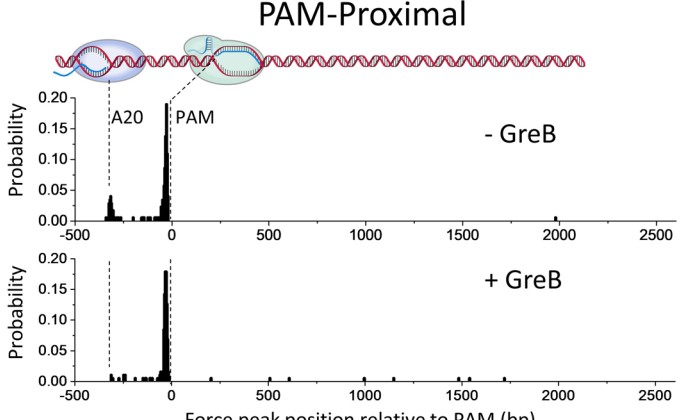

## PAM-Proximal

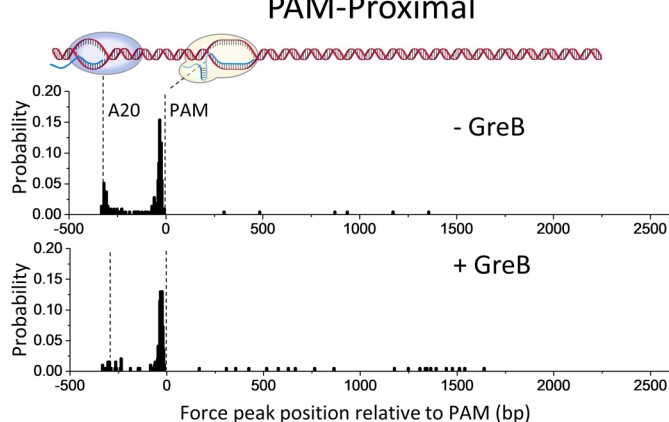

**Extended Data Fig. 3 | RNAP locations upon collisions with a bound dCas protein for data shown in Fig. 2.** Distributions of RNAP force peak locations after NTP addition for dCas9 (**a**) or dCas12a (**b**) for each condition in Fig. 2. RNAP locations were determined after quenching transcription assays with a PAM-distal (top) or PAM-proximal (bottom) bound dCas complex. Each dCas orientation was assayed either in the presence or the absence of GreB during chasing. The expected locations of the A20 and the PAM site are indicated as dashed lines. The RNAP force peak locations were pooled for $N$ biologically independent traces: dCas9 PAM-distal, $N = 217$ (−GreB) and $N = 184$ (+GreB); dCas9 PAM-proximal, $N = 174$ (−GreB) and $N = 190$ (+GreB); dCas12a PAM-distal, $N = 160$ (−GreB) and $N = 177$ (+ GreB); dCas12a PAM-proximal, $N = 214$ (−GreB) and $N = 192$ (+GreB). Source Data File for all histograms is provided.

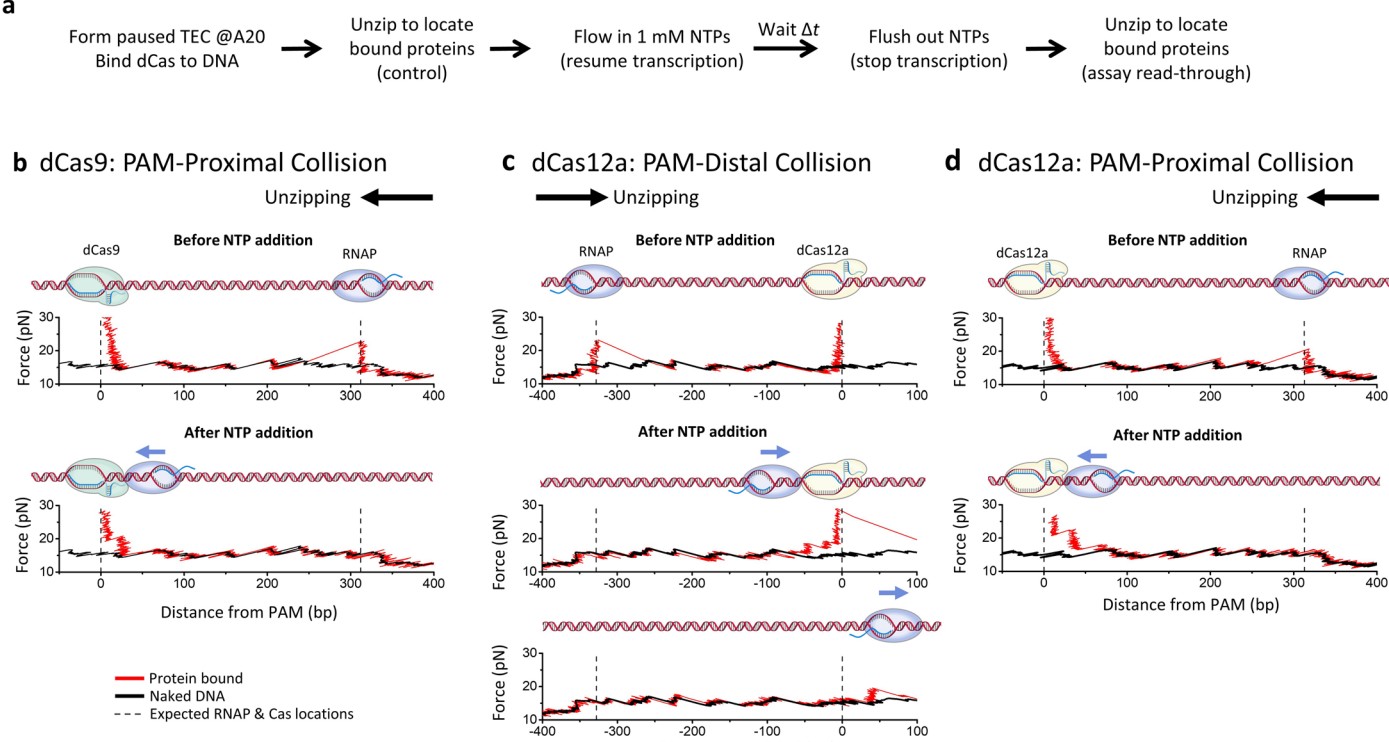

**Extended Data Fig. 4 | Representative traces of transcription encountering dCas9 from the PAM proximal side and encountering dCas12a from both sides. a**, Flowchart of the transcription read-through assay. **b**, Representative traces of RNAP encountering a bound dCas9 from the PAM proximal side. An example control trace is shown with RNAP and bound dCas9 detected at their expected locations. After NTP addition, shown are example traces of RNAP prior to encountering the dCas9 (top) and RNAP colliding with dCas9 (bottom). **c**, Representative traces of transcription assays with dCas12a in the PAM-Distal orientation. An example control trace is shown with RNAP and

bound dCas12a detected at their expected locations. After NTP addition, shown are example traces of RNAP prior to encountering the dCas12a (top), RNAP colliding with dCas12a (middle), and RNAP reading through dCas12a (bottom). **d**, Representative traces of RNAP encountering a bound dCas12a from the PAM proximal side. An example control trace is shown with RNAP and bound dCas12a detected at their expected locations. After NTP addition, shown are example traces of RNAP prior to encountering the dCas12a (top) and RNAP colliding with dCas12a (bottom). Source Data File for plots in b-d is provided.

**Extended Data Fig. 5 | Trace classification for transcription assays.** Cartoons represent the observed states in a sample chamber before chase (left) and after chase (right). Arrows indicate possible transitions between initial and final states. This diagram informs equations that represent different pathways for transitions between the initial and final states as described in methods and is used to solve for the relevant parameters of read-through and dCas removal.

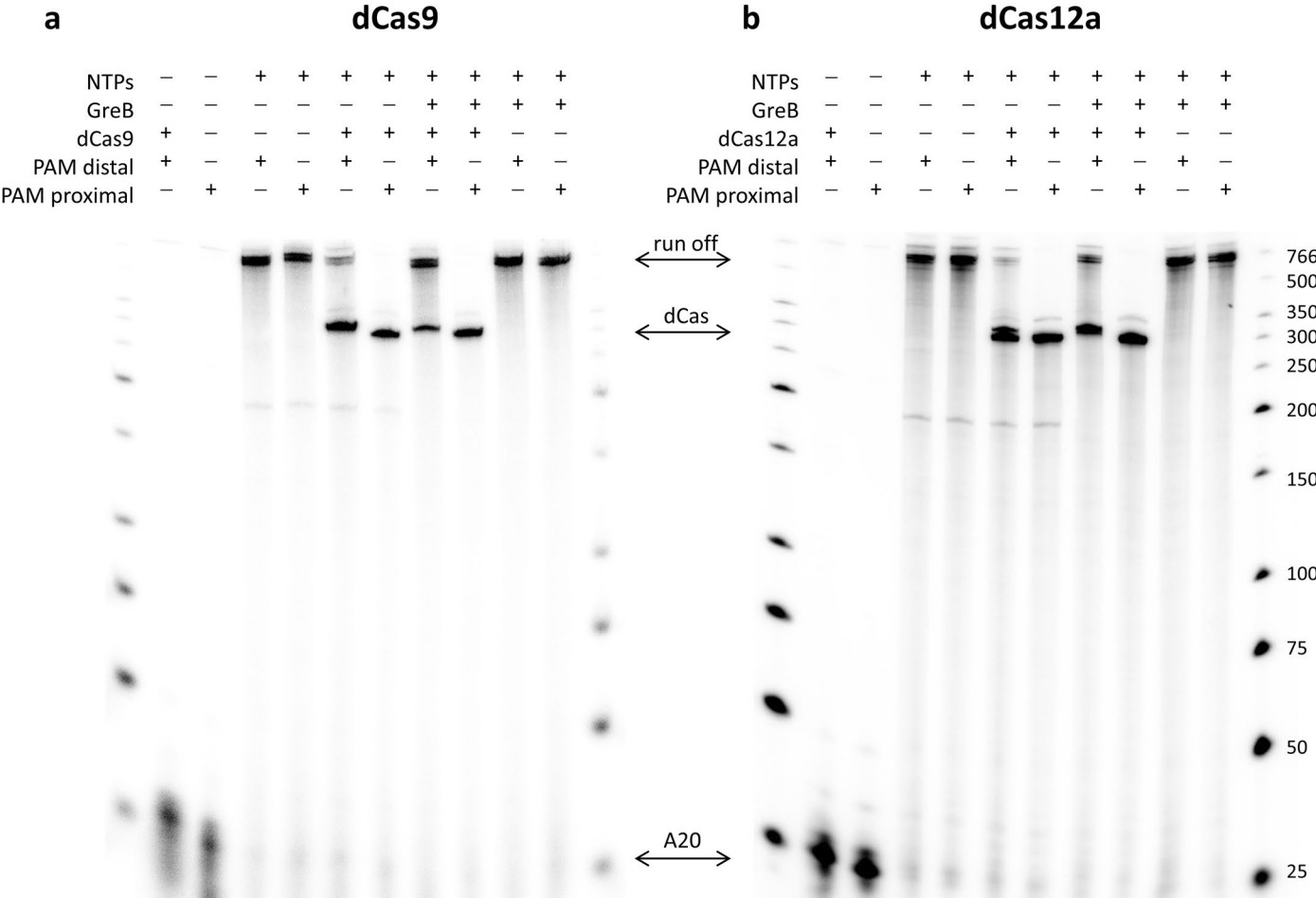

**Extended Data Fig. 6 | Read-through polarity of dCas9 and dCas12a via bulk transcription.** Shown are results from bulk transcription run-off assays with bound dCas9 (**a**) or dCas12a (**b**) proteins in either the PAM-distal or PAM-proximal orientation relative to the promoter. Transcription was carried out with 1 mM NTPs in the presence or absence of 1 μM GreB for 135 s before being quenched with formamide and EDTA to stop the reaction. Transcripts were assayed by 6% denaturing PAGE gels (Methods). The distance from the +1 to the PAM sequence for the PAM-distal orientation collision was 349 bp for both dCas9 and dCas12a, while this distance for the PAM-proximal collision was 332 bp for dCas9 and 333 bp for dCas12a (Supplementary Table 5). The locations of the A20, dCas collision, and run-off products are indicated with arrows. The transcription read-throughs from these gels were ~35% without GreB and ~77% with GreB for PAM-distal dCas9 collisions, ~6% without GreB and ~6% with GreB for PAM-proximal dCas9 collisions, ~31% without GreB and ~54% with GreB for PAM-distal dCas12a collisions, and ~9% without GreB and ~8% with GreB for PAM-distal dCas12a collisions. We performed an additional transcription gel assay for each dCas complex, and those gels yielded similar results as shown here. Source Data File containing uncropped gel scans is provided.

**a**

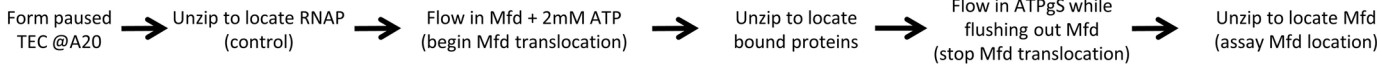

Form paused TEC @A20 → Unzip to locate RNAP (control) → Flow in Mfd + 2mM ATP (begin Mfd translocation) → Unzip to locate bound proteins → Flow in ATPgS while flushing out Mfd (stop Mfd translocation) → Unzip to locate Mfd (assay Mfd location)

**b**

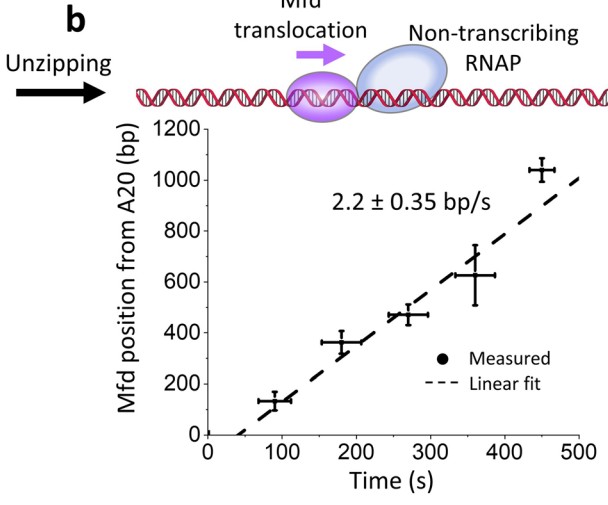

Unzipping

Mfd translocation

Non-transcribing RNAP

2.2 ± 0.35 bp/s

- Measured
--- Linear fit

**c**

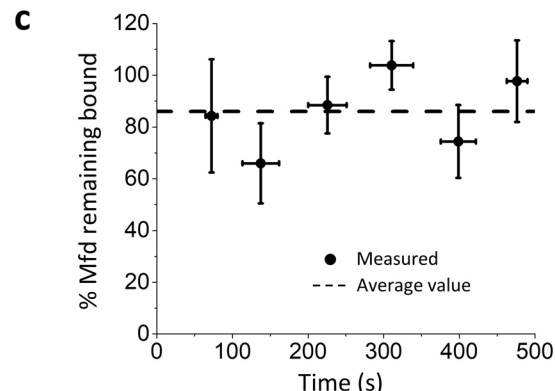

- Measured
--- Average value

**Extended Data Fig. 7 | Mfd speed and processivity. a**, Flowchart of the Mfd only control experiments. Sample chambers containing TEC paused at 20 are formed as in Figs. 2–5, and dCas was not bound the DNA. In contrast to Fig. 3, tethers were unzipped while Mfd was translocating to assess Mfd moving in 'real-time'. **b**, Mfd translocation versus time. $N$ number of biologically independent traces were used for analysis at each time point: $N = 2$ at 0 s, $N = 10$ at 90 s, $N = 21$ at 180 s, $N = 25$ at 270 s, $N = 11$ at 360 s, and $N = 6$ at 450 s. Error bars are SEMs for the vertical axis and SDs for the horizontal axis. The black dash line is a linear fit, giving a speed of 2.2 ± 0.35 bp/s. **c**, % Mfd remaining bound versus time. Each data point is normalized against the fraction of tethers initially containing an Mfd. $N$ number of biologically independent traces were used for analysis at each time point: $N = 5$ at 72 s, $N = 25$ at 137 s, $N = 30$ at 225 s, $N = 31$ at 310 s, $N = 31$ at 398 s, $N = 15$ at 476 s. Error bars are SEMs. Source Data File for b and c is provided.

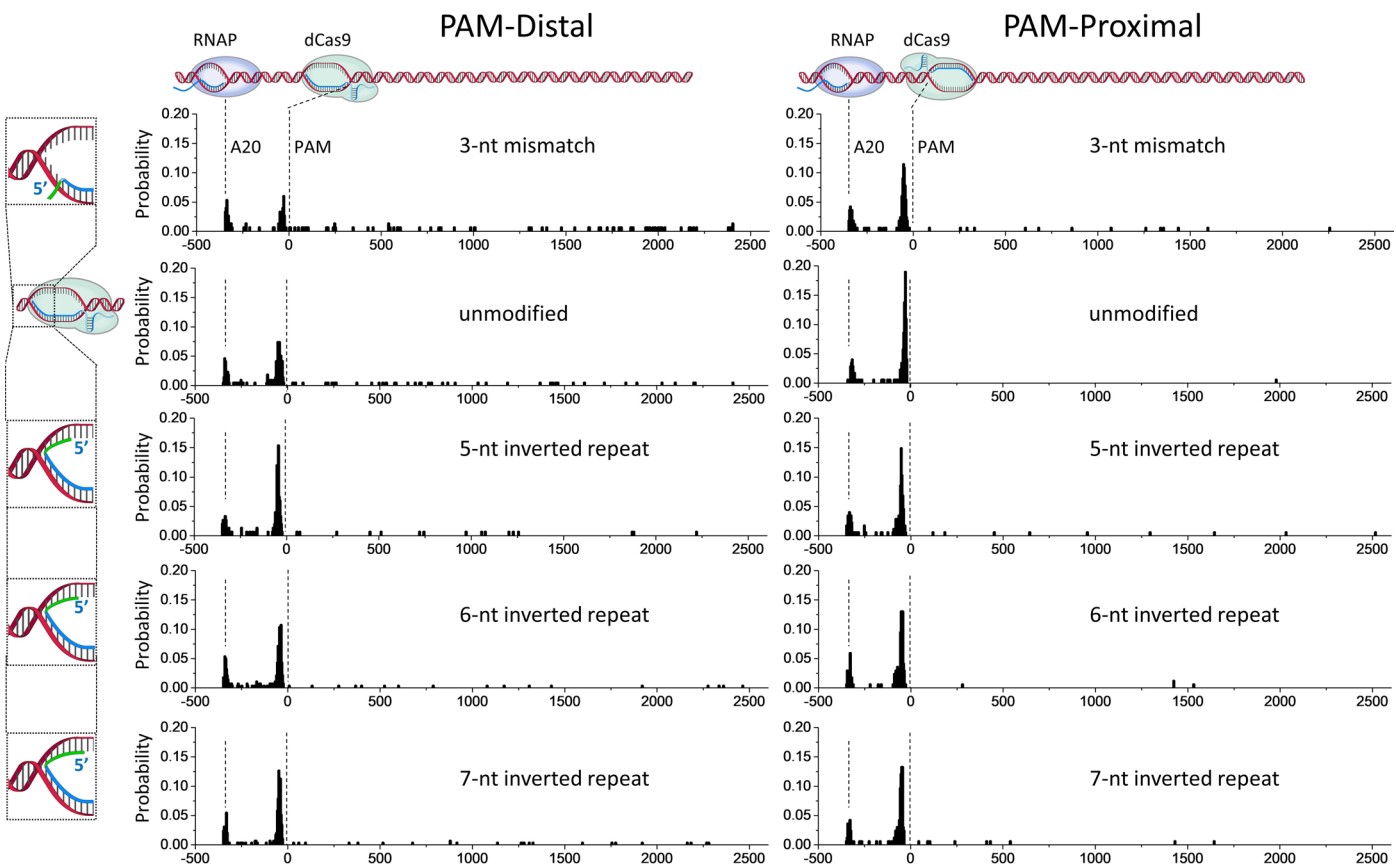

**Extended Data Fig. 8 | RNAP locations upon collisions with a bound dCas protein for data shown in Figs. 4, 5.** Distributions of RNAP force peak locations after NTP addition for each condition in Figs. 4, 5. The locations of the A20 and the PAM site are indicated with dashed lines. RNAP locations were determined after quenching transcription assays with a PAM-distal (left) or PAM-proximal (right) bound dCas complex. The RNAP force peak locations were pooled for $N$ biologically independent traces: dCas9 PAM-distal, $N = 150$ (3-nt mismatch), $N = 217$ (unmodified), $N = 151$ (5-nt IR), $N = 282$ (6-nt IR), $N = 295$ (7-nt IR); dCas9 PAM-proximal, $N = 165$ (3-nt mismatch), $N = 174$ (unmodified), $N = 175$ (5-nt IR), $N = 169$ (6-nt IR), and $N = 167$ (7-nt IR). Source Data File for b and c is provided.

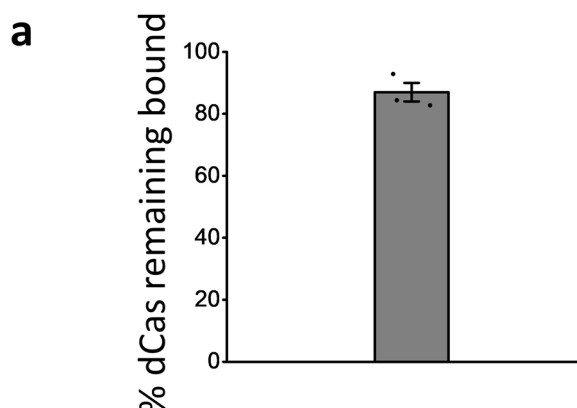

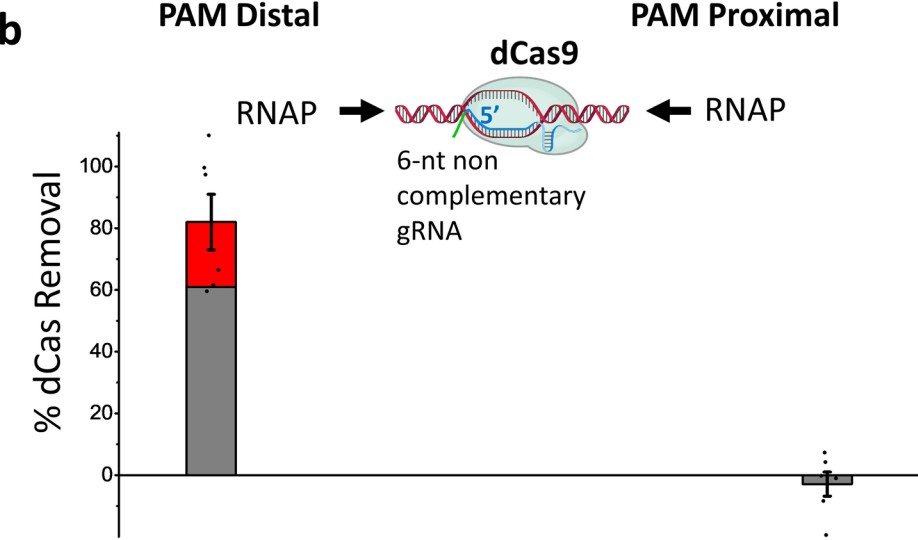

**Extended Data Fig. 9 | Transcription read-through and removal efficiency of a bound dCas9 complexed with gRNA containing a 6-nt extension, which is not complementary to the target DNA. a**, % of dCas remaining bound after the quench. A control experiment to measure the fraction of dCas9 containing this modified gRNA (Supplementary Table 2) remaining bound after the introduction of the transcription conditions, using the method described in Extended Data Fig. 2d. For each sample chamber, both control traces and non-control traces were taken to obtain the % dCas remaining for that chamber. The experiment was repeated using $N = 3$ biologically independent sample chambers. The % dCas remaining value was calculated for each sample chamber (black dots), and the mean value and SEM of these repeats are also shown. **b**, The transcription assay was performed using the modified gRNA for both the PAM-distal and PAM-proximal orientations. For each sample chamber, both control traces and non-control traces were taken to obtain the % dCas removal for that chamber. The experiment was repeated using $N = 6$ biologically independent sample chambers. The % dCas removal value was calculated for each sample chamber (black dots), and the mean value and SEM of these repeats are also shown. Source Data File for a and b is provided.

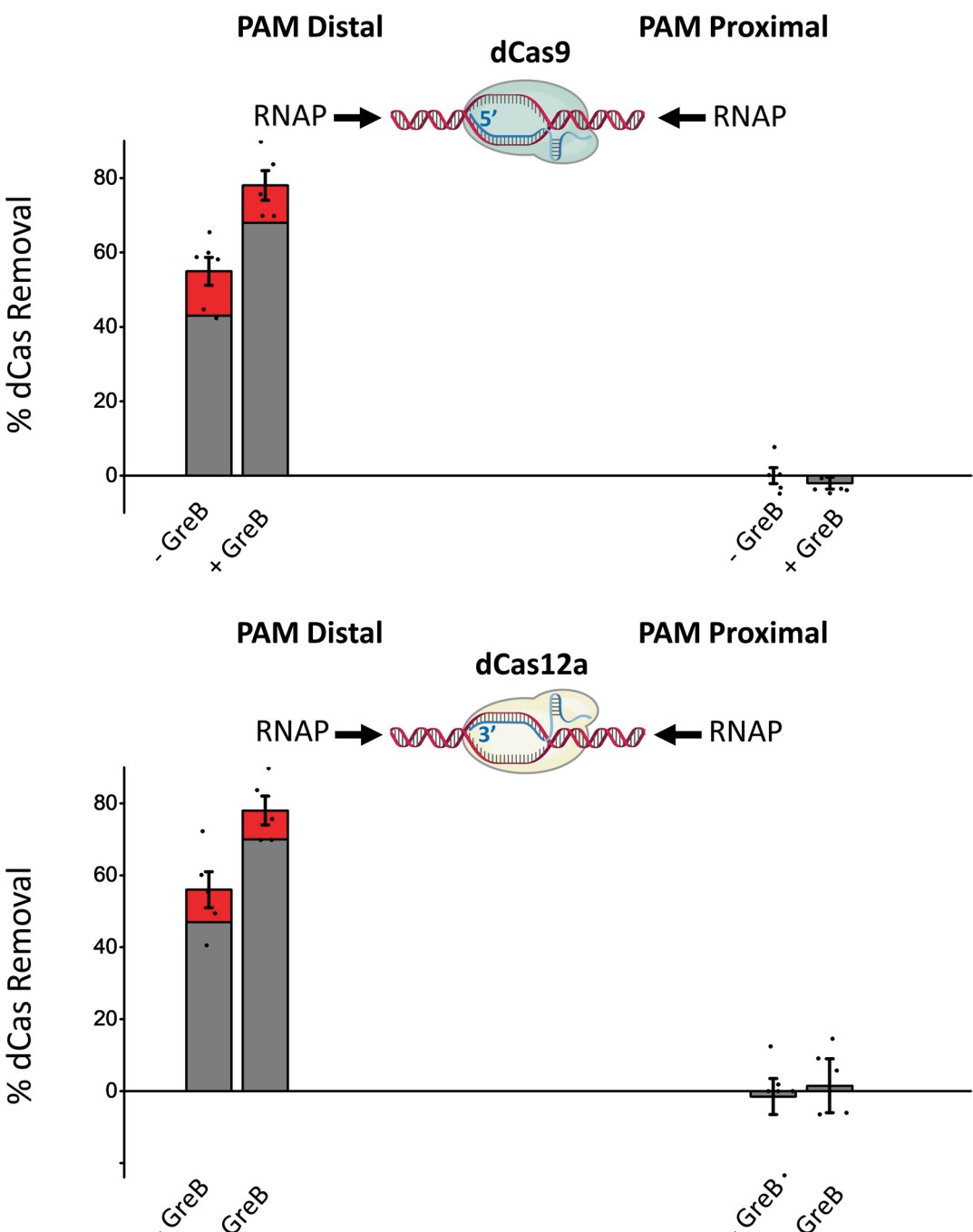

**Extended Data Fig. 10 | dCas9 and dCas12a removal efficiency during transcription in the presence of GreB.** The same data taken for Fig. 2c were used for this analysis, so the sample trace statistics are identical to those for Fig. 2c. Out of the removal efficiency, the fraction from traces with removal due to transcription read-through (grey) and the fraction from traces with removal but without transcription read-through (red) are stacked. Black dots are removal efficiencies calculated for each independent replicate. Error bars are SEMs. Source Data File for these bar graphs is provided.

# Reporting Summary

## Statistics

For all statistical analyses, confirm that the following items are present in the figure legend, table legend, main text, or Methods section.

| n/a | Confirmed | |
|---|---|---|
| ☐ | ☒ | The exact sample size ($n$) for each experimental group/condition, given as a discrete number and unit of measurement |
| ☐ | ☒ | A statement on whether measurements were taken from distinct samples or whether the same sample was measured repeatedly |
| ☒ | ☐ | The statistical test(s) used AND whether they are one- or two-sided<br>*Only common tests should be described solely by name; describe more complex techniques in the Methods section.* |
| ☒ | ☐ | A description of all covariates tested |
| ☒ | ☐ | A description of any assumptions or corrections, such as tests of normality and adjustment for multiple comparisons |
| ☐ | ☒ | A full description of the statistical parameters including central tendency (e.g. means) or other basic estimates (e.g. regression coefficient) AND variation (e.g. standard deviation) or associated estimates of uncertainty (e.g. confidence intervals) |
| ☒ | ☐ | For null hypothesis testing, the test statistic (e.g. $F$, $t$, $r$) with confidence intervals, effect sizes, degrees of freedom and $P$ value noted<br>*Give P values as exact values whenever suitable.* |
| ☒ | ☐ | For Bayesian analysis, information on the choice of priors and Markov chain Monte Carlo settings |
| ☒ | ☐ | For hierarchical and complex designs, identification of the appropriate level for tests and full reporting of outcomes |
| ☒ | ☐ | Estimates of effect sizes (e.g. Cohen's $d$, Pearson's $r$), indicating how they were calculated |

*Our web collection on statistics for biologists contains articles on many of the points above.*

## Software and code

Policy information about availability of computer code

| Data collection | LabView 2017 optical tweezers software |
|---|---|
| Data analysis | Matlab 2017 optical tweezers trace analysis software |

For manuscripts utilizing custom algorithms or software that are central to the research but not yet described in published literature, software must be made available to editors and reviewers. We strongly encourage code deposition in a community repository (e.g. GitHub). See the Nature Portfolio guidelines for submitting code & software for further information.

## Data

Policy information about availability of data

All manuscripts must include a data availability statement. This statement should provide the following information, where applicable:
- Accession codes, unique identifiers, or web links for publicly available datasets
- A description of any restrictions on data availability
- For clinical datasets or third party data, please ensure that the statement adheres to our policy

Relevant source data for main text and supplementary figures are provided in the Source Data section. All other data that support the findings of this study are available from the corresponding author upon reasonable request.

# Field-specific reporting

Please select the one below that is the best fit for your research. If you are not sure, read the appropriate sections before making your selection.

☒ Life sciences ☐ Behavioural & social sciences ☐ Ecological, evolutionary & environmental sciences

For a reference copy of the document with all sections, see nature.com/documents/nr-reporting-summary-flat.pdf

# Life sciences study design

All studies must disclose on these points even when the disclosure is negative.

| | |
|---|---|
| Sample size | The sample size for each condition was 5-8 biologically independent sample chambers. Each sample chamber represents a unique assembly of proteins onto multiple DNA substrates. Note that within each sample chamber, multiple traces (on the order of 50 traces, see Source Data Files), each on a different DNA molecule, were measured to characterize the presence and location of proteins on the DNA. No sample size calculation was performed. The sample size was determined based on our experience with single-molecule measurements in order to accommodate variations in protein-DNA binding efficiency. |
| Data exclusions | No data were excluded from the analysis unless there were DNA tethering issues (e.g., DNA broke during measurement). |
| Replication | All the experiments described in the manuscript were reproducible. |
| Randomization | Randomization was not relevant to this study, as all experiments were carried out using the same experimental procedures. |
| Blinding | Blinding was not relevant to this study. All data collection was done following the same standard procedures. |

# Reporting for specific materials, systems and methods

We require information from authors about some types of materials, experimental systems and methods used in many studies. Here, indicate whether each material, system or method listed is relevant to your study. If you are not sure if a list item applies to your research, read the appropriate section before selecting a response.

## Materials & experimental systems

| n/a | Involved in the study |
|---|---|
| ☐ | ☒ Antibodies |
| ☒ | ☐ Eukaryotic cell lines |
| ☒ | ☐ Palaeontology and archaeology |
| ☒ | ☐ Animals and other organisms |
| ☒ | ☐ Human research participants |
| ☒ | ☐ Clinical data |
| ☒ | ☐ Dual use research of concern |

## Methods

| n/a | Involved in the study |
|---|---|
| ☒ | ☐ ChIP-seq |
| ☒ | ☐ Flow cytometry |
| ☒ | ☐ MRI-based neuroimaging |

## Antibodies

| | |
|---|---|
| Antibodies used | Goat anti-digoxygenin, Vector Labs, #MB-7000, lot ZD0727 |
| Validation | Anti-digoxygenin was used to tether DNA to the surface. It was validated by the observation of DNA labeled with digoxygenin tethered to the surface. The specificity and quality of the anti-body does not affect the outcome of the single-molecule study. No additional data are necessary to validate the antibody. |

