## [Peer Review File · Nature Structural & Molecular Biology]

Peer Review Information

Journal: Nature Structural and Molecular Biology

Manuscript Title: Polarity of the CRISPR Roadblock to Transcription

Corresponding author name(s): Professor Michelle Wang

Editorial Notes:

Redactions – unpublished data	Parts of this Peer Review File have been redacted as indicated to maintain the confidentiality of unpublished data.
Redactions – confidential patient information	Parts of this Peer Review File have been redacted as indicated to maintain patient confidentiality.
Redactions – published data	Parts of this Peer Review File have been redacted as indicated to remove third-party material.
Redactions – reviewer opt-out	Parts of this Peer Review File have been redacted as indicated as we could not obtain permission to publish the reports of reviewer no. XX .
Reviewer comments in marked-up manuscript	In their review of the [first/second/third/...] version of this manuscript, reviewer no. XX added their comments to the manuscript file. These comments, excluding minor textual revisions, have been copied into this Peer Review File.

Reviewer Comments & Decisions:

Decision Letter, initial version:

1st Mar 2022

Dear Dr. Wang,

Thank you again for submitting your manuscript "Polarity of the CRISPR Roadblock to Transcription". I apologize for the delay while we awaited the comments (below) from the 3 reviewers who evaluated

your paper. In light of their reports, we remain interested in your study and would like to see your response to the comments of the referees, in the form of a revised manuscript.

You will see that Reviewers 1 and 3 are positive about the quality of the data and potential interest of the findings, and that Reviewer 3 suggests minor revisions of text and figures. Reviewer 1 finds that the novel aspects of the study would be strengthened by additional experimentation to determine how force variation correlates with transcriptional activity. Editorially, we agree that these data would strengthen the work, and ask that they be included in a revised manuscript. While we appreciate Reviewer 2's request for additional *in vivo* analyses, we do not feel these assays are essential to support the central conclusions of the present work and would not ask that they be included. We also only consider peer-reviewed publications when gauging the novel insights provided by a manuscript submitted to our journal.

Please be sure to address/respond to all concerns of the referees in full in a point-by-point response and highlight all changes in the revised manuscript text file. If you have comments that are intended for editors only, please include those in a separate cover letter.

When revising the manuscript, please bear in mind the following guidelines for our Article format:

- abstract should be maximum 150 words, no references;
- main text is typically between 3,000 and 4,000 words, and should be organized as introduction, results (with subheadings) and discussion.
- display items (figures and tables): typically between 6 and 8. Please note that the structural table should be in main article.
- supplementary items: Supplementary Figures should be a maximum of 10; other supplementary items are Suppl Table, Note, Video, Data Set.

We expect to see your revised manuscript within 6 weeks. If you cannot send it within this time, please contact us to discuss an extension; we would still consider your revision, provided that no similar work has been accepted for publication at NSMB or published elsewhere.

Reporting Summary:

Data availability: this journal strongly supports public availability of data. All data used in accepted papers should be available via a public data repository, or alternatively, as Supplementary Information. If data can only be shared on request, please explain why in your Data Availability Statement, and also in the correspondence with your editor. Please note that for some data types, deposition in a public repository is mandatory - more information on our data deposition policies and available repositories can be found below:
<https://www.nature.com/nature-research/editorial-policies/reporting-standards#availability-of-data>

[Redacted]

With kind regards,

Beth

Beth Moorefield, Ph.D.
Senior Editor
Nature Structural & Molecular Biology

Referee expertise:

Referee #1: transcription/single-molecule studies

Referee #2: self identifies: CRISPR function/structural biology

Referee #3: transcription/structural biology

Reviewers' Comments:

Reviewer #1:

Remarks to the Author:

The manuscript titled "Polarity of the CRISPR Roadblock to Transcription" by Hall et al reports on the asymmetric Cas binding stability to target DNA which depends on the PAM-proximal vs. PAM-distal orientation. Cas binding to target DNA results in R-loop formation which can be removed by RNA polymerase. Previous studies already showed that RNAP is blocked by dCas from the PAM-proximal but not from the PAM-distal side, which revealed the asymmetric stability of dCas binding. The current study seeks to gain a quantitative understanding of this asymmetric roadblock effect by applying dCas9 and dCas12a in conjunction with RNAP to their DNA unzipping mapper technique. They demonstrate that dCas R-loop allows RNAP to read through only from the PAM-distal side, not from the PAM-proximal side, which matches the previous report. Further, the addition of GreB and Mfd still showed the strong polar barrier effect of dCas9 and 12a, suggesting the tightly engaged state of the PAM-proximal side of both Cas systems. This is a rigorous study which is clearly written with beautiful figures that are easy to follow along. The novelty is somewhat weak in that the asymmetric nature of dCas with respect to RNAP was already known. However, the current findings add relative force parameters to the previous observations. One key experiment that should be performed and presented in the manuscript is the transcription measurement. This can simply be done in bulk solution in the presence of the same template and same set of proteins as applied to the single molecule assay. PAM-proximal should not produce any RNA, but PAM-distal should produce transcript. It will be exciting to see if the force variation is correlated to the transcriptional output. This will reveal the barrier force-function relationship that connects to biological systems. Overall, this is a nice piece of work but as it stands, it's stands as a confirmation of the previous finding. Below are some additional points to address.

1. DNA unzipping dimension should be present at least in the beginning figure of the paper. As drawn, it is somewhat dubious as to what force is being applied to the system.
2. The authors state that there are two types of traces obtained for dCas12a. Both types of traces need to be shown for the main figure and they need to be quantified. What is the fraction of type I vs. type II? Are they always distinguishable?
3. Overall, the authors need to provide how many traces were used for the analysis.
4. "transcription read-through of a bound dCas9 showed an efficiency of 43% when RNAP approached dCas9 from the PAM-distal side." In this experiment, what is the distribution of the transcription read-through? In other words, how far do RNAPs travel beyond dCas9 position?
5. Related to above, such distribution should be shown for other conditions of varying percentage of

RNAP read through.

6. It is difficult to understand why the dCas blockade effect is identical between the template vs. non-template since RNAP tracks along the template while non template is displaced away.

7. How do we know if GreB bound to both PAM-distal and PAM-proximal side? Gel?

8. GreB effect on PAM-distal will be much more convincing if shown by transcription output i.e longer product and/or higher product generation.

9. The authors do not talk about the GC vs. AT rich sequence composition at the PAM-distal vs. -proximal gRNA. It will be practically helpful to know what role sequence plays at these two ends.

10. How do we know the inverted repeat sequence gets tucked into the DNA duplex as shown in the figure? It can be that the extended sequence makes the barrier effect to be greater.

Reviewer #2:

Remarks to the Author:

Hall et al. use an optical tweezer trap assay to determine the effect of catalytically dead Cas9 and Cas12 on the progression of RNAP and Mfd. As far as I can tell, the assays have been performed with a high level of technical expertise. Given the wealth of existing structural information, it is unclear what conceptual advances this study provides. This reviewer does not recommend publication in NSMB. The work appears to be more suitable for a specialized journal such as Biophysical J.

Major points:

1. The conclusion that RNAP and Mfd read-through is abolished when Cas9 is approached from the PAM-proximal but not the PAM-distal end is not at all surprising. The PAM-proximal end has far more interactions with Cas9 than the PAM-distal end does, and is therefore more stable. This has been demonstrated in numerous structural studies dating back to 2014. What have you learned that is new based on your studies? Are there more advances that this reviewer is missing?

2. Furthermore, three recent preprints (Pacesa et al., 2021, Cofsky et al., 2021, Bravo et al., 2021) have shown that at various stages of R-loop formation (and completion) the PAM-distal end of the gRNA:TS duplex is flexible, whereas the PAM-proximal region is not. Again, what have you learned that provides a conceptual advance?

3. It would be interesting if you could show that this has consequences for either editing or recruitment of DNA repair factors in vivo. This would be a clear advance and be of broad interest to general readers. Have you tried performing such experiments? Even if this could be done using your assay, it would be important.

David Taylor

Reviewer #3:

Remarks to the Author:

Hall et al. apply high-resolution single-molecule optical trap technique (DNA unzipping) to confirm previously observed polarity of dCas barrier to transcription and, for the first time, reveal the mechanism of this polarity. Specifically, dCas interactions in the R-loop (PAM-distal) region are weaker and so the R-loop is collapsed by the approaching RNA polymerase vs. stronger interactions of dCas with the PAM-proximal region of DNA preclude forward movement of RNA polymerase. This mechanism extends to other motor proteins, like Mfd translocase. Besides these qualitative observations, the authors do a thorough job quantifying transcription/translocation read through the polar dCas roadblock. Furthermore, the authors use a clever strategy to increase/decrease R-loop stability with inverted repeats/mismatches at 5' end of guide RNA, which allows modulation of transcription levels through dCas barrier. Overall, this work is well thought out, with proper controls, and has immediate implications for CRISPRi and in vivo pulldown experiments. It will be exciting to see how inverted repeat gRNAs will behave in vivo. All in all, it deserves the visibility of publication in NSMB.

I only have a few minor points for the authors to address:

"Previous studies" are cited throughout. Could the authors briefly mention what experimental methods, orthogonal to the ones used in the present work, were employed? E.g., first paragraph on p. 5.

Figures: could you please explicitly mark "force drop" areas to make them easier to spot?

Last paragraph of Discussion on p. 15: could the authors elaborate on what "other gRNA designs and/or Cas protein mutagenesis that stabilize the R-loop" might be? This will be of interest to CRISPRi community.

Missing "of" on p. 14 second paragraph: "This finding may be interpreted in light of our mechanism of CRISPRi polarity."

Missing "refs" and a typo ("pre-purposed") at the bottom of p. 14

Author Rebuttal to Initial comments
--

Response to Reviewers

Re: NSMB-A45720
“Polarity of the CRISPR Roadblock to Transcription” by Hall et al.

We greatly appreciate the helpful comments and suggestions from the three reviewers and the Editor. We have carefully considered each of the comments and have taken comprehensive actions to address them. We believe that these changes further strengthen the original conclusions and enhance the clarity of the manuscript.

Below, we begin with a summary of changes made to the manuscript. This is followed by a detailed, point-by-point reply to each comment from the three reviewers. Each comment (bold) is followed by our reply (not bold).

Summary of Changes:

In both the main text and SI, we have highlighted changes in red. Below, we summarize the most significant changes.

1. As requested by Reviewer #1, we have conducted bulk transcription assays to determine transcription read through of a bound Cas9 or Cas12a protein, both in the absence or presence of GreB. These results are now shown as the new Supplementary Fig. 6 with their protocols discussed in the expanded Methods section. These new results support and further substantiate our single-molecule unzipping data.
2. In response to comments from Reviewer #1, we have conducted new single-molecule experiments to investigate whether an extended sequence of a gRNA may strengthen a bound dCas9 roadblock when RNAP collides with the dCas9 from the PAM-distal side. We found that the extended gRNA actually weakens the dCas9 roadblock. These new data are shown in Supplementary Fig. 9 and provide further evidence that our modified gRNAs with an inverted repeat create an extended R-loop that resists removal by RNAP.
3. As requested by Reviewer #1, we have plotted the locations of the detected RNA polymerase (RNAP) for the data shown in Fig. 2 and Fig. 4. These analyses are now shown as the new Supplementary Fig. 3 and new Supplementary Fig. 8, respectively.
4. As requested by Reviewer #1 and Reviewer #3, we have revised Fig. 1 to 1) include a cartoon of the experimental configuration, 2) show the second configuration of a bound Cas12a when unzipped from the PAM distal side, and 3) indicate locations for the force dips.

Reviewer #1:

The manuscript titled “Polarity of the CRISPR Roadblock to Transcription” by Hall et al reports on the asymmetric Cas binding stability to target DNA which depends on the PAM-proximal vs. PAM-distal orientation. Cas binding to target DNA results in R-loop formation which can be removed by RNA polymerase. Previous studies already showed that RNAP is blocked by dCas from the PAM-proximal

but not from the PAM-distal side, which revealed the asymmetric stability of dCas binding. The current study seeks to gain a quantitative understanding of this asymmetric roadblock effect by applying dCas9 and dCas12a in conjunction with RNAP to their DNA unzipping mapper technique. They demonstrate that dCas R-loop allows RNAP to read through only from the PAM-distal side, not from the PAM-proximal side, which matches the previous report. Further, the addition of GreB and Mfd still showed the strong polar barrier effect of dCas9 and 12a, suggesting the tightly engaged state of the PAM-proximal side of both Cas systems. This is a rigorous study which is clearly written with beautiful figures that are easy to follow along. The novelty is somewhat weak in that the asymmetric nature of dCas with respect to RNAP was already known. However, the current findings add relative force parameters to the previous observations. One key experiment that should be performed and presented in the manuscript is the transcription measurement. This can simply be done in bulk solution in the presence of the same template and same set of proteins as applied to the single molecule assay. PAM-proximal should not produce any RNA, but PAM-distal should produce transcript. It will be exciting to see if the force variation is correlated to the transcriptional output. This will reveal the barrier force-function relationship that connects to biological systems. Overall, this is a nice piece of work but as it stands, it stands as a confirmation of the previous finding. Below are some additional points to address.

We appreciate the critical reading from Reviewer #1 and a chance to clarify the results of the manuscript. We agree that it would be interesting to also perform parallel bulk transcription experiments. As discussed below, we have performed these experiments, and the findings from these new data are consistent with those of our single-molecule data.

As you will see from our response to Reviewer #2, although previous studies have shown that a bound dCas is a polar roadblock to transcription, what governs the polarity remained unclear. Curiously, previous studies also show that a bound dCas is not a polar roadblock to replication. Thus, the polarity is specific to the type of motor protein and thus should be dictated by dynamics of both the motor protein and the bound dCas. This might have not been clear in the initial manuscript, and we have clarified this on pages 2 and 14-15 in the revised manuscript.

What is really novel about our work is that our proposed mechanism explains both the presence of polarity for transcription and the absence of polarity for replication. RNAP is able to *rezip* its downstream DNA as it moves forward, and in doing so, squeezes out the RNA in the R-loop of a bound dCas protein and thus destabilizes the bound dCas protein. A replisome, however, uses its helicase to *unzip* the DNA to separate the two DNA strands downstream and therefore cannot rezip DNA to collapse the R-loop of a bound dCas complex. To our knowledge, this is the first mechanistic explanation of these apparently disparate findings of dCas roadblock polarity for transcription and replication. Understanding the nature of the polarity has important implications for CRISPR design and applications *in vivo*.

1. DNA unzipping dimension should be present at least in the beginning figure of the paper. As drawn, it is somewhat dubious as to what force is being applied to the system.

We thank Reviewer #1 for this suggestion. We have moved an experimental configuration cartoon from the SI to a main figure. It is now the new Fig. 1a. This cartoon shows how an optical trap is used to unzip a DNA molecule and where forces are applied on the DNA unzipping template. We hope that this cartoon clarifies how our unzipping experiments are conducted.

2. The authors state that there are two types of traces obtained for dCas12a. Both types of traces need to be shown for the main figure and they need to be quantified. What is the fraction of type I vs. type II? Are they always distinguishable?

Following the suggestion from Reviewer #1, we have moved the second type of trace of dCas12a from the SI to the main figure. It is now part of Fig. 1b, which shows both types of trace. The second type is distinguished from the first type by having another force peak near the distal end of the gRNA/DNA hybrid. We have also provided the statistics of each type of traces on page 4.

3. Overall, the authors need to provide how many traces were used for the analysis.

We have revised the legends of Figs. 2-5 and Supplementary Figs. 2 and 7-9 to provide detailed statistics of each experimental condition on the total number of the sample chambers, the numbers of control traces, and the numbers of non-control traces.

4. “transcription read-through of a bound dCas9 showed an efficiency of 43% when RNAP approached dCas9 from the PAM-distal side.” In this experiment, what is the distribution of the transcription read-through? In other words, how far do RNAPs travel beyond dCas9 position?

Following the suggestion from Reviewer #1, we have plotted the RNAP locations for all conditions used in Fig. 2 and these data are now in the new Supplementary Fig. 3. As shown in this new figure, when RNAP approached dCas9 or dCas12a from the PAM-distal side, a fraction of RNAP travelled beyond the bound dCas protein with their positions distributed across the DNA template, and this fraction increased in the presence of GreB. For completeness, we also show RNAP locations when RNAP approached a bound dCas9 or dCas12a from the PAM-proximal side, where minimal RNAP read-through was detected.

5. Related to above, such distribution should be shown for other conditions of varying percentage of RNAP read through.

We have now also shown RNAP locations for conditions using the various modified gRNAs of Figs. 4 and 5 in the new Supplementary Fig. 8.

6. It is difficult to understand why the dCas blockade effect is identical between the template vs. non-template since RNAP tracks along the template while non template is displaced away.

As the Reviewer noted, the roadblock polarity of a bound dCas protein depends on PAM-distal or PAM proximal encounter, not on whether the gRNA is on the template or non-template strand (see Fig. 1c). Although RNAP tracks along the template strand, RNAP tightly clamps down and re-zips dsDNA downstream as it moves forward. Because the template and non-template strands are brought together in front of the RNAP, RNAP's front edge is insensitive to the strand the gRNA of a bound dCas protein is on, when it is encountered from the PAM-distal side. RNAP's front edge holds onto both strands of the DNA and re-zips the DNA within the R-loop to destabilize the bound dCas. This tight grip of RNAP on its downstream dsDNA is also highlighted in Fig. 1c. We hope that this clarifies this point.

7. How do we know if GreB bound to both PAM-distal and PAM-proximal side? Gel?

Following the suggestion from this Reviewer, we have performed bulk transcription assays for transcription through dCas9 or dCas12a from either the PAM distal side or PAM proximal side, in the

presence or absence of GreB. These data are now shown as the new Supplementary Fig. 6. These data show that for both dCas9 or dCas12a, the presence of GreB increased read-through from the PAM distal side. Although the presence of GreB did not increase the read through from the PAM-proximal side, we could tell that it was active in the reactions by the disappearance of a weak pause site located prior to RNAP encountering the abundant dCas protein (at a transcript size of ~ 180 nt).

8. GreB effect on PAM-distal will be much more convincing if shown by transcription output i.e longer product and/or higher product generation.

This comment is related to the one above. As shown by the transcription gels (now the new Supplementary Fig. 6) as well as single-molecule data (the new Supplementary Fig. 3), the presence of GreB indeed has led to longer products and higher product generation for the PAM-distal side.

9. The authors do not talk about the GC vs. AT rich sequence composition at the PAM-distal vs. - proximal gRNA. It will be practically helpful to know what role sequence plays at these two ends.

Supplementary Table 2 shows the full sequences of crRNAs used in this work. As shown in this table, the crRNA used have no particular bias of GC or AT at either the PAM-distal or PAM-proximal end.

10. How do we know the inverted repeat sequence gets tucked into the DNA duplex as shown in the figure? It can be that the extended sequence makes the barrier effect to be greater.

We very much appreciate this comment from Reviewer #1. In the original manuscript, we showed that a gRNA containing an extended sequence with an inverted repeat is a stronger roadblock to RNAP from the PAM-distal side (Fig. 4). We interpreted this strengthened roadblock as a consequence of the gRNA straddling the two strands of the DNA bubble and, as the Reviewer puts it, being tucked into the DNA duplex. Fig. 4b indeed supports this interpretation. The Reviewer raises the possibility of another interpretation – that the extended gRNA sequence itself may give rise to an increased barrier effect.

We took this comment from the Reviewer to heart and performed a new set of experiments. For these experiments, we designed the gRNA with an extension that is *not* complementary to the original gRNA so that the extended sequence cannot hybridize with the DNA bubble (and thus it cannot be tucked into the DNA duplex). Interestingly, we observed significant transcription read-through from the PAM-distal side, comparable to that using the 3-nt mismatched gRNA and consistent with the extended sequence actually decreasing the barrier effect. The extended non-complementary sequence may dangle away from the R-loop of a bound dCas9, facilitating the start of gRNA-DNA separation during RNAP invasion into the DNA bubble.

Thus, these control experiments strengthen our initial interpretation. These new gRNA data are now discussed on page 11, and shown in the new Supplementary Fig. 9.

Reviewer #2:

Hall et al. use an optical tweezer trap assay to determine the effect of catalytically dead Cas9 and Cas12 on the progression of RNAP and Mfd. As far as I can tell, the assays have been performed with a high level of technical expertise. Given the wealth of existing structural information, it is unclear what

conceptual advances this study provides. This reviewer does not recommend publication in NSMB. The work appears to be more suitable for a specialized journal such as Biophysical J.

We appreciate this Reviewer for taking the time to comment on our manuscript.

Major points:

1. The conclusion that RNAP and Mfd read-through is abolished when Cas9 is approached from the PAM-proximal but not the PAM-distal end is not at all surprising. The PAM-proximal end has far more interactions with Cas9 than the PAM-distal end does, and is therefore more stable. This has been demonstrated in numerous structural studies dating back to 2014. What have you learned that is new based on your studies? Are there more advances that this reviewer is missing?

2. Furthermore, three recent preprints (Pacesa et al., 2021, Cofsky et al., 2021, Bravo et al., 2021) have shown that at various stages of R-loop formation (and completion) the PAM-distal end of the gRNA:TS duplex is flexible, whereas the PAM-proximal region is not. Again, what have you learned that provides a conceptual advance?

Since the first two comments are related, we will address them together. The Reviewer is correct that there have been many structural studies on DNA-bound Cas9 or Cas12a, which show that the PAM-proximal end has more interactions with Cas proteins than the PAM-distal end, and the gRNA/DNA duplex of the PAM-distal end is more flexible than the gRNA/DNA duplex of the PAM-proximal end. In our original manuscript, we cited some of those publications and have now added another one (Cofsky et al., 2020). However, this structural information alone does not explain the roadblock polarity for transcription. To overcome a dCas roadblock, RNAP must overcome *all* interactions within a bound dCas protein, regardless of whether those interactions are located on the PAM-proximal side or the PAM-distal side. If the dCas roadblock strength solely depends on the sum of the interactions within a bound dCas protein, then the dCas roadblock should have no polarity.

Importantly, as we discussed in the manuscript, previous studies have shown that a bound dCas is a polar roadblock to transcription but *not* to replication. Thus, the polarity is specific to the type of motor protein and thus should be dictated by dynamics of both the motor protein and the bound dCas. We do not think that any structural data alone have provided any explanations for these observations.

What is really novel about our work is that our proposed mechanism explains both the presence of polarity for transcription and the absence of polarity for replication. RNAP is able to *rezip* its downstream DNA as it moves forward, and in doing so, squeezes out the RNA in the R-loop of a bound dCas protein and thus destabilizes the bound dCas protein. A replisome, however, uses its helicase to *unzip* the DNA to separate the two DNA strands downstream and therefore cannot rezip DNA to collapse the R-loop of a bound dCas complex. To our knowledge, this is the first mechanistic explanation of these apparently disparate findings of dCas roadblock polarity for transcription and replication. Understanding the nature of the polarity has important implications for CRISPR design and applications *in vivo*.

We hope that this explanation provides some clarification on the novelty of the work.

3. It would be interesting if you could show that this has consequences for either editing or recruitment of DNA repair factors in vivo. This would be a clear advance and be of broad interest to general readers. Have you tried performing such experiments? Even if this could be done using your assay, it would be important.

We agree that it would be very interesting to show that our findings have consequences for either editing or recruitment of DNA repair factors *in vivo*. Unfortunately, those are beyond the scope of the current work.

Reviewer #3:

Hall et al. apply high-resolution single-molecule optical trap technique (DNA unzipping) to confirm previously observed polarity of dCas barrier to transcription and, for the first time, reveal the mechanism of this polarity. Specifically, dCas interactions in the R-loop (PAM-distal) region are weaker and so the R-loop is collapsed by the approaching RNA polymerase vs. stronger interactions of dCas with the PAM-proximal region of DNA preclude forward movement of RNA polymerase. This mechanism extends to other motor proteins, like Mfd translocase. Besides these qualitative observations, the authors do a thorough job quantifying transcription/translocation read through the polar dCas roadblock. Furthermore, the authors use a clever strategy to increase/decrease R-loop stability with inverted repeats/mismatches at 5' end of guide RNA, which allows modulation of transcription levels through dCas barrier. Overall, this work is well thought out, with proper controls, and has immediate implications for CRISPRi and *in vivo* pulldown experiments. It will be exciting to see how inverted repeat gRNAs will behave *in vivo*. All in all, it deserves the visibility of publication in NSMB.

We thank the Reviewer for giving our manuscript a careful read and for providing positive and encouraging comments.

I only have a few minor points for the authors to address:

“Previous studies” are cited throughout. Could the authors briefly mention what experimental methods, orthogonal to the ones used in the present work, were employed? E.g., first paragraph on p. 5.

We have clarified the methods used in “previous studies” in the first paragraph on pages 4, 5, and 7.

Figures: could you please explicitly mark “force drop” areas to make them easier to spot?

We thank the Reviewer for this suggestion. In Fig. 1b, we now use red arrows to specifically indicate where the force dips below the naked DNA baseline. These arrows should help the readers focus on regions that are particularly relevant.

Last paragraph of Discussion on p. 15: could the authors elaborate on what “other gRNA designs and/or Cas protein mutagenesis that stabilize the R-loop” might be? This will be of interest to CRISPRi community.

We appreciate this comment from the Reviewer. Our work has demonstrated two avenues that impact Cas binding – *stability* of the R-loop and *access* to the R-loop. We imagine that this information may open the door for new approaches to modulate Cas binding.

For example, one could exploit the energetics of a gRNA-DNA hybrid by minimizing hairpin formation within an inverted repeat gRNA while optimizing the stability of the extended R-loop. Although this requires thermodynamically favorable sequence locations, this optimization may not significantly limit utility given the immense sequence space in genomes of interest.

As another example, fusing Cas with a DNA binding protein may restrict access to the R loop. Upon Cas binding, the DNA binding protein could bind the dsDNA that flanks the PAM-distal end of the R-loop, protecting that region of dsDNA and thus making it harder for RNAP to invade and disrupt the bound Cas.

To describe these ideas succinctly, we have provided pared-down examples and slightly modified the paragraph on Page 16.

Missing “of” on p. 14 second paragraph: “This finding may be interpreted in light of our mechanism of CRISPRi polarity.”

We thank the Reviewer for spotting this typo. This sentence has now been replaced.

Missing “refs” and a typo (“pre-purposed”) at the bottom of p. 14.

We thank the Reviewer for spotting these issues, which have now been corrected.

Decision Letter, first revision:

Our ref: NSMB-A45720A

9th Sep 2022

Dear Dr. Wang,

Thank you for submitting your revised manuscript "Polarity of the CRISPR Roadblock to Transcription" (NSMB-A45720A). I again sincerely apologize for the delay in getting back to you, which resulted from difficulties in reaching one of the referees. It has now finally been seen by two of the original reviewers and their comments are below. They find that the paper has improved in revision, and therefore we'll be happy in principle to publish it in Nature Structural & Molecular Biology, pending minor revisions to comply with our editorial and formatting guidelines.

To facilitate our work at this stage, we would appreciate if you could send us the main text as a word file. Please make sure to copy the NSMB account (cc'ed above).

Kind regards,
Florian

Dr Florian Ullrich
Associate Editor, Nature
Consulting Editor, Nature Structural & Molecular Biology
ORCID 0000-0002-1153-2040

Reviewer #1 (Remarks to the Author):

The authors revised their manuscript thoroughly by performing additional experiments requested by the reviewers. The manuscript is greatly strengthened and the mechanism is much better supported. It is ready for publication.

Reviewer #3 (Remarks to the Author):

I am happy with the manuscript in its current form and appreciate the authors addressing the points I raised in the original submission.

Decision Letter, final checks:

Our ref: NSMB-A45720A

22nd Sep 2022

Dear Dr. Wang,

Thank you for your patience as we've prepared the guidelines for final submission of your Nature Structural & Molecular Biology manuscript, "Polarity of the CRISPR Roadblock to Transcription" (NSMB-A45720A). Please carefully follow the step-by-step instructions provided in the attached file, and add a response in each row of the table to indicate the changes that you have made. Please also check and comment on any additional marked-up edits we have proposed within the text. Ensuring that each point is addressed will help to ensure that your revised manuscript can be swiftly handed over to our production team.

In recognition of the time and expertise our reviewers provide to Nature Structural & Molecular Biology's editorial process, we would like to formally acknowledge their contribution to the external peer review of your manuscript entitled "Polarity of the CRISPR Roadblock to Transcription". For those reviewers who give their assent, we will be publishing their names alongside the published article.

Nature Structural & Molecular Biology offers a Transparent Peer Review option for new original research manuscripts submitted after December 1st, 2019. As part of this initiative, we encourage our authors to support increased transparency into the peer review process by agreeing to have the reviewer comments, author rebuttal letters, and editorial decision letters published as a Supplementary item. When you submit your final files please clearly state in your cover letter whether or not you would like to participate in this initiative. Please note that failure to state your preference will result in delays in accepting your manuscript for publication.

Cover suggestions

As you prepare your final files we encourage you to consider whether you have any images or

illustrations that may be appropriate for use on the cover of Nature Structural & Molecular Biology.

Nature Structural & Molecular Biology has now transitioned to a unified Rights Collection system which will allow our Author Services team to quickly and easily collect the rights and permissions required to publish your work. Approximately 10 days after your paper is formally accepted, you will receive an email in providing you with a link to complete the grant of rights. If your paper is eligible for Open Access, our Author Services team will also be in touch regarding any additional information that may be required to arrange payment for your article.

Please note that *Nature Structural & Molecular Biology* is a Transformative Journal (TJ). Authors may publish their research with us through the traditional subscription access route or make their paper immediately open access through payment of an article-processing charge (APC). Authors will not be required to make a final decision about access to their article until it has been accepted. [Find out more about Transformative Journals](https://www.springernature.com/gp/open-research/transformative-journals)

Authors may need to take specific actions to achieve [compliance with funder and institutional open access mandates](https://www.springernature.com/gp/open-research/funding/policy-compliance-faqs). If your research is supported by a funder that requires immediate open access (e.g. according to [Plan S principles](https://www.springernature.com/gp/open-research/plan-s-compliance)) then you should select the gold OA route, and we will direct you to the compliant route where possible. For authors selecting the subscription publication route, the journal's standard licensing terms will need to be accepted, including [self-archiving policies](https://www.nature.com/nature-portfolio/editorial-policies/self-archiving-and-license-to-publish). Those licensing terms will supersede any other terms that the author or any third party may assert apply to any version of the manuscript.

For information regarding our different publishing models please see our [Transformative Journals](https://www.springernature.com/gp/open-research/transformative-journals) page. If you have any questions about costs, Open Access requirements, or our legal

forms, please contact ASJournals@springernature.com.

[Redacted]

Best regards,

Sophia Frank
Editorial Assistant
Nature Structural & Molecular Biology
nsmb@us.nature.com

On behalf of

Florian Ullrich, Ph.D.
Associate Editor
Nature Structural & Molecular Biology
ORCID 0000-0002-1153-2040

Reviewer #1:

Remarks to the Author:

The authors revised their manuscript thoroughly by performing additional experiments requested by the reviewers. The manuscript is greatly strengthened and the mechanism is much better supported. It is ready for publication.

Reviewer #3:

Remarks to the Author:

I am happy with the manuscript in its current form and appreciate the authors addressing the points I raised in the original submission.

Author Rebuttal, First Revision:

Response to Reviewers

Re: NSMB-A45720A
"Polarity of the CRISPR Roadblock to Transcription" by Hall et al.

We greatly appreciate the time and effort of Reviewer #1 and Reviewer #3 for re-reviewing our manuscript and the Editors for securing these reviewers. Both reviewers are satisfied with our manuscript (see their comments below). They do not have any comments that require our response.

Reviewer #1:

The authors revised their manuscript thoroughly by performing additional experiments requested by the reviewers. The manuscript is greatly strengthened and the mechanism is much better supported. It is ready for publication.

Reviewer #3:

I am happy with the manuscript in its current form and appreciate the authors addressing the points I raised in the original submission.

Final Decision Letter:

12th Oct 2022

Dear Dr. Wang,

We are now happy to accept your revised paper "Polarity of the CRISPR Roadblock to Transcription" for publication as a Article in Nature Structural & Molecular Biology.

As soon as your article is published, you can generate your shareable link by entering the DOI of your article here: <http://authors.springernature.com/share>.

Corresponding authors will also receive an automated email with the shareable link

Note the policy of the journal on data deposition:

<http://www.nature.com/authors/policies/availability.html>.

Your paper will be published online soon after we receive proof corrections and will appear in print in the next available issue. You can find out your date of online publication by contacting the production team shortly after sending your proof corrections. Content is published online weekly on Mondays and

Thursdays, and the embargo is set at 16:00 London time (GMT)/11:00 am US Eastern time (EST) on the day of publication. Now is the time to inform your Public Relations or Press Office about your paper, as they might be interested in promoting its publication. This will allow them time to prepare an accurate and satisfactory press release. Include your manuscript tracking number (NSMB-A45720B) and our journal name, which they will need when they contact our press office.

About one week before your paper is published online, we shall be distributing a press release to news organizations worldwide, which may very well include details of your work. We are happy for your institution or funding agency to prepare its own press release, but it must mention the embargo date and Nature Structural & Molecular Biology. If you or your Press Office have any enquiries in the meantime, please contact press@nature.com.

Please note that *Nature Structural & Molecular Biology* is a Transformative Journal (TJ). Authors may publish their research with us through the traditional subscription access route or make their paper immediately open access through payment of an article-processing charge (APC). Authors will not be required to make a final decision about access to their article until it has been accepted. [Find out more about Transformative Journals](https://www.springernature.com/gp/open-research/transformative-journals)

Authors may need to take specific actions to achieve [compliance](https://www.springernature.com/gp/open-research/funding/policy-compliance-faqs) with funder and institutional open access mandates. If your research is supported by a funder that requires immediate open access (e.g. according to [Plan S principles](https://www.springernature.com/gp/open-research/plan-s-compliance)) then you should select the gold OA route, and we will direct you to the compliant route where possible. For authors selecting the subscription publication route, the journal's standard licensing terms will need to be accepted, including [13](https://www.springernature.com/gp/open-

self-archiving policies. Those licensing terms will supersede any other terms that the author or any third party may assert apply to any version of the manuscript.

Kind regards,
Florian

Dr Florian Ullrich
Associate Editor, Nature
Consulting Editor, Nature Structural & Molecular Biology
ORCID 0000-0002-1153-2040